Unlocking Andean sigmodontine diversity: five new species of Chilomys (Rodentia: Cricetidae) from the montane forests of Ecuador

http://orcid.org/0000-0002-3410-6669 Brito Jorge 1 jorgeyakuma@yahoo.es
http://orcid.org/0000-0002-2196-1199 Tinoco Nicolás 2
http://orcid.org/0000-0002-3640-2357 Pinto C. Miguel 3
García Rubí 1
http://orcid.org/0000-0002-7115-2816 Koch Claudia 4
http://orcid.org/0000-0002-8315-1458 Fernandez Vincent 5
Burneo Santiago 2
http://orcid.org/0000-0001-9496-5433 Pardiñas Ulyses F. J. 1 6
1 Sección de Mastozoología, Instituto Nacional de Biodiversidad (INABIO) , Quito, Pichincha , Ecuador
2 Sección de Mastozoología, Museo de Zoología, Facultad de Ciencias Exactas y Naturales, Pontificia Universidad Católica del Ecuador , Quito, Pichincha , Ecuador
3 Observatorio de Biodiversidad Ambiente y Salud (OBBAS), Quito, Pichincha, Ecuador. Current address: Charles Darwin Research Station, Charles Darwin Foundation , Puerto Ayora, Galápagos , Ecuador
4 Leibniz Institute for the Analysis of Biodiversity Change/Museum Koenig , Bonn , Germany
5 Imaging and Analysis Centre, Natural History Museum (NHM) , London , United Kingdom
6 Instituto de Diversidad y Evolución Austral (IDEAus – CONICET) , Puerto Madryn, Chubut , Argentina
Morrone Juan J.
Electronic publication date: 2022 Apr 19
Publication date: 2022
Volume: 10
Electronic Location ID: e13211
Received 2021 Dec 15; Accepted 2022 Mar 11
Copyright: © 2022 Brito et al.
Copyright year: 2022
Copyright holder: Brito et al.
License: This is an open access article distributed under the terms of the Creative Commons Attribution License, which permits unrestricted use, distribution, reproduction and adaptation in any medium and for any purpose provided that it is properly attributed. For attribution, the original author(s), title, publication source (PeerJ) and either DOI or URL of the article must be cited.
License URL: https://creativecommons.org/licenses/by/4.0/

Keywords: Andes, CT, Proodonty, Microdonty, Thomasomyini, Sigmodontinae

Funding: Fundación EcoMinga Germany-Brazil-Ecuador Trilateral Cooperation Program, funded by the International Cooperation GIZ Secretaría de Educación Superior, Ciencia Tecnología e Innovación (SENESCYT) This work was supported by Fundación EcoMinga (Jorge Brito and Ulyses F. J. Pardiñas), Germany-Brazil-Ecuador Trilateral Cooperation Program, funded by the international cooperation GIZ (Jorge Brito and Claudia Koch). The laboratory work in Ecuador was funded by grants from Secretaría de Educación Superior, Ciencia Tecnología e Innovación (SENESCYT), Project ‘Arca de Noe’, S. R. Ron and O. Torres-Carvajal Principal Investigators (N. Tinoco). The funders had no role in study design, data collection and analysis, decision to publish, or preparation of the manuscript.

==============================
The Andean cloud forests of Ecuador are home to several endemic mammals. Members of the Thomasomyini rodents are well represented in the Andes, with Thomasomys being the largest genus (47 species) of the subfamily Sigmodontinae. Within this tribe, however, there are genera that have escaped a taxonomic revision, and Chilomys Thomas, 1897, constitutes a paradigmatic example of these “forgotten” Andean cricetids. Described more than a century ago, current knowledge of this externally unmistakable montane rodent is very limited, and doubts persist as to whether or not it is monotypic. After several years of field efforts in Ecuador, a considerable quantity of specimens of Chilomys were collected from various localities representing both Andean chains. Based on an extensive genetic survey of the obtained material, we can demonstrate that what is currently treated as C. instans in Ecuador is a complex comprising at least five new species which are described in this paper. In addition, based on these noteworthy new evidence, we amend the generic diagnosis in detail, adding several key craniodental traits such as incisor procumbency and microdonty. These results indicate that Chilomys probably has a hidden additional diversity in large parts of the Colombian and Peruvian territories, inviting a necessary revision of the entire genus.

Introduction

Our current understanding of Andean sigmodontine rodents is mostly driven by the noticeable diversity of the genera Calomys, Phyllotis and Thomasomys. Clearly, they are emblematic widespread and speciose taxa, Thomasomys representing the largest part of the subfamily with 47 species (see Brito et al., 2021; Ruelas & Pacheco, 2021), and received extensive attention covering aspects from alpha taxonomy (e.g., Pearson, 1957; Hershkovitz, 1962; Zeballos et al., 2014; Salazar-Bravo, 2015; Steppan & Ramirez, 2015; Pacheco, 2015a; Martínez, Sandoval & Carrizo, 2016) to physiology, reproduction, etc. (e.g., Arana et al., 2002; Tirado, Cortés & Bozinovic, 2008; Brito & Batallas, 2014; Sahley et al., 2015, 2016). In the Andes, however, several other sigmodontine genera exist that are much less studied and are considered paucispecific, such as Aepeomys, Chilomys, Galenomys, or Neomicroxus. These taxa, characterized by being poorly represented in biological collections (e.g., Galenomys; Pearson, 1957) and sometime considered rare (e.g., Aepeomys; Handley, 1976), have traditionally escaped systematic revisions. Nevertheless, they constitute a substantial expression of Andean sigmodontine diversity, particularly in northern South America, and have the potential to expand our current comprehension of cricetid evolution in this complex part of the continent (e.g., Soriano et al., 1999; Voss, 2003; Anderson et al., 2012; Cañón et al., 2020).

Chilomys Thomas, 1897, constitutes a paradigmatic example of these ‘forgotten’ Andean cricetids. Described more than a century ago, our current knowledge of this externally unmistakable montane rodent is very scarce (Thomas, 1895; Osgood, 1912; Pacheco, 2015a; Brito & Pardiñas, 2017). Although this genus was considered monotypic for most of its history, it now consists of two speciesm C. fumeus Osgood, 1912, restricted to the northernmost Andes in Colombia and Venezuela, and the widespread C. instans (Thomas, 1895), the type species of the genus, which occurs from central Colombia to northern Perú (Medina et al., 2016). Both forms are considered very similar (in fact, they have been largely considered synonyms, see Musser & Carleton, 2005) and were distinguished by subtle metric characters (Pacheco, 2015b: 578). But the existence of possible undescribed species has also been suggested for Colombian (Pacheco, 2015b: 580), Ecuadorian (Pinto et al., 2018: 18) and Peruvian populations (Medina et al., 2016: 317).

After several years of field efforts in Ecuador, researchers have collected a considerable quantity of specimens of Chilomys from various localities representing both Andean chains. These populational samples allowed surpassing a traditional impediment in the systematic revision of this genus: the scarcity of available material to assess variability (Voss, 2003). Based on an extensive genetical survey of the obtained material, we can demonstrate that what is currently understood as C. instans in Ecuador is a complex comprising at least five new species. The purpose of the present contribution is to document these findings to initiate a much-needed revision of the entire genus.

Materials and Methods

Studied specimens

This study implies a qualitative and metrical revision based on 97 specimens belonging to the genus Chilomys from populations in Ecuador and, subsidiarily, Colombia (Appendix 1). Most of the Ecuadorian specimens studied were collected by the senior author and collaborators during recent field trips conducted in the Cordillera de Kutukú, Reserva Drácula, Parque Nacional Sangay, the Cordillera de Chilla, Reserva Geobotánica Pululahua and Reserva Naturetrek Vizcaya. These surveys involved a cumulative trap effort of 12,800 trap/nights. Capture, handling and preservation of specimens secured in the field followed established guidelines of the American Society of Mammalogists (Sikes, 2016). Research permits were obtained from the Ministry of the Environment of Ecuador ((scientific research authorization No 006-2015-IC-FLO-FAU-DPAC/MAE, 003-2019-ICFLO-FAU-DPAC/MAE), MAE-DNB-CM-2019-0126, and MAAE-ARSFC-2020-0642). The collected material was compared with specimens housed in the mammal collections of the following institutions: Centro Nacional Patagónico, Puerto Madryn, Chubut, Argentina (CNP); Instituto Nacional de Biodiversidad, Quito, Ecuador (MECN; formerly known as Museo Ecuatoriano de Ciencias Naturales); Museo de la Escuela Politécnica Nacional, Quito, Ecuador (MEPN); Museo de Zoología de la Pontificia Universidad Católica del Ecuador, Quito, Ecuador (QCAZ); Field Museum of Natural History, Chicago, USA (FMNH); and the Natural History Museum, London, United Kingdom (NHMUK).

Anatomy, age criteria and measurements

Terms used to describe cranial anatomy follow Carleton & Musser (1989), Musser et al. (1998), Pacheco (2003), and Voss (1993); occlusal molar morphology are based on Reig (1977) with upper and lower molars identified as M/m, respectively. The description of the coloration is made based on Köhler (2012). Soft anatomy is assessed according to the concepts discussed by Carleton (1973) and Vorontsov (1982) on stomach and caecum, by Vorontsov (1982) and Voss (1988) on tongue, by Quay (1954) on soft palate, by Ade (1999) and Haidarliu, Kleinfeld & Ahissar (2013) on rhinarium, and by Pacheco (2003) on anus. Terminology and definitions follow Tribe (1996) and Costa et al. (2011) for age classes, and the term “adults” is restricted to individuals categorized as age three and four. External measurements (always provided in millimetres, mm), were mostly recorded in the field and derive from specimens tags; these descriptors are: head and body length (HBL), tail length (TL), hind foot length (HF, including claw), ear length (E), and body mass (W, in grams). Cranial measurements were obtained with a digital calliper to the nearest 0.01 mm, and include the following dimensions (see Tribe, 1996; Voss, 2003; and Musser et al., 1998, for definitions and illustrations): condylo-incisive length (CIL), condylo-basal length (CBL), zygomatic breadth (ZB), least interorbital breadth (LIB), length of rostrum (LR), breadth of rostrum (BR), length of nasals (LN), length of upper diastema (LD), crown length of maxillary toothrow (LM), length of incisive foramina (LIF), breadth of incisive foramina (BIF), breadth of bony palate (BPB), breadth of mesopterygoid fossa (BM), depth of upper incisor (DI), breadth of zygomatic plate (BZP), braincase breadth (BB), breadth of Ml (BM1), breadth of the incisor tips (BIT), length of mandible (LMN), crown length of mandibular toothrow (LLM), and depth of mandibular ramus (DR).

X-ray micro CT

For more detailed analysis and representation of the morphological characteristics of the skulls, several specimens selected as holotypes (MECN 5854, MECN 6024) or paratype (MECN 3723) of the new species described herein were scanned using a high-resolution X-ray micro-computed tomography desktop device (micro-CT; Bruker SkyScan 1173, Kontich, Belgium) at the Leibniz Institute for the Analysis of Biodiversity Change/Museum Koenig (LIB, Bonn, Germany). To avoid movements during scanning, the skulls were embedded in cotton wool and placed in a small plastic container. Acquisition parameters comprised: An X-ray beam (source voltage 30 kV and current 170 µA) without the use of a filter; 800 (MECN 3723, MECN 6024) to 1,200 (MECN 5854) projections of 900 ms exposure time each with a frame averaging of 6 (MECN 3723, MECN 6024) to 7 (MECN 5854); rotation steps of 0.2° (MECN 5854) to 0.3° (MECN 3723, MECN 6024) recorded over a 180° continuous rotation, resulting in a scan duration of 1 h 36 min (MECN 3723, MECN 6024) to 2 h 43 min (MECN 5854); and a magnification setup generating data with an isotropic voxel size of 14.55 µm (MECN 3723), 13.48 µm (MECN 5854) and 13.84 µm (MECN 6024), respectively. The CT-datasets were reconstructed with N-Recon software version 1.7.1.6 (Bruker MicroCT, Kontich, Belgium) and rendered in three dimensions using CTVox for Windows 64 bits version 3.0.0 r1114 (Bruker MicroCT, Kontich, Belgium). For comparison, the holotype of Chilomys instans (NHMUK 1985.10.14.1) was scanned at the Imaging Analysis Centre of the NHMUK using a Nikon Metrology XTH 225 ST (Nikon Metrology, Leuven, Belgium). Acquisition parameters comprised: an X-ray beam (source voltage 85 kV and current 118 µA) filtered with 0.1 mm of aluminium; 4,476 projections of 250 ms exposure time each with a frame averaging of 2 recorded over a 360° continuous rotation; a magnification setup generating data with an isotropic voxel size of 11.57 µm. A filtered back projection algorithm was used for the tomographic reconstruction, using the CT-agent and CT-pro 3D software (Version 6; Nikon Metrology, Tring, United Kingdom), producing an 8-bit uncompressed raw volume. Finally, this dataset was rendered in three dimensions with Amira software (Thermo Fisher Scientific, Hillsboro, OR, USA).

Morphometric analyses

The analyzed dataset of craniodental measurements comprised 21 variables, from 58 specimens belonging to six taxa, including typical Chilomys instans and the five new species described here (for the analysis we used adult and old specimens). We performed all subsequent analyses in R version 3.6.2 (R Core Team, 2019), unless otherwise noted. We tested each measurement for normality using the Shapiro-Wilk test using the R function shapiro.test. Four of these measurements were not normally distributed; thus we transformed the whole dataset, using the R function log, to improve its statistical properties. The dataset had 1% of missing data so, to avoid eliminating individuals or measurements from the analyses, we performed imputation of missing data in the R implementation of the program Amelia II (Honaker, King & Blackwell, 2011) with the expectation-maximization (EM) method because of its higher accuracy (Strauss, Atanassov & de Oliveira, 2003; Clavel, Merceron & Escargue, 2014). We generated 100 imputed datasets (m = 100), which we averaged to obtain a single imputed dataset using the Python script avg.py (Mark, 2017). Prior to these analyses we checked for unusually high pair-wise correlations among measurements using the R function cor. The variables CIL, CBL and LD were highly correlated (r > 0.95), so we removed the variables CBL and LD from the multivariate analyses restricting the final dataset to 19 variables. We conducted two multivariate analyses: a principal component analysis (PCA) with the covariance matrix using the R function princomp, and a discriminant function analysis (DFA) using the R script MorphoTools version 1.1 (Koutecký, 2015). We drew the scatter plots of the PCA and DFA with the R function plot.

DNA amplification and sequencing

We used samples of liver and muscle tissues (preserved in 95% ethanol) and in some cases fragments of dry skin. We extracted DNA using the salt protocol (Bilton & Jaarola, 1996), and amplified by PCR two mitochondrial genes. The first gene was Cytochrome b (Cytb) since it has been widely used for studies of taxonomic revisions in different sigmodontine rodents (e.g., Smith & Patton, 1999; Bonvicino & Moreira, 2001; Hanson et al., 2015; Brandão et al., 2021; Saldanha & Rossi, 2021). The second gene was Cytochrome Oxidase I (COI) is a gene that also allows to detect intraspecific variations and possible new species (e.g., Pinto et al., 2018; Andrade et al., 2021). For Cytb we used the primers MVZ05, MVZ16H and MVZ14 (Smith & Patton, 1993) and thermal protocols reported by Smith & Patton (1999) and Bonvicino & Moreira (2001). We PCR amplified the COI gene using the cocktail of primers for mammals and the thermal protocol reported by Ivanova et al. (2007). PCR protocols are specified in Supplemental S1. We visually evaluated the quality of the PCR amplicons with gel electrophoresis and subsequently we purified the amplicons with Exosap-IT (GE Healthcare, Chalfont St. Giles, UK). Macrogen Inc. (Seoul, South Korea) sequenced the PCR amplicons with Sanger technology.

Phylogenetic analysis

In Geneious R11 (https://www.geneious.com) we assembled and edited the sequences and aligned them using the ClustalW tool. We obtained the best partition schemes and respective models of evolution with PartitionFinder V.1 (Lanfear et al., 2017): for the Cytb gene: 1pos GTR + I + G, 2pos HKY + G, 3pos + I + G; and for COI gene the first, second and third positions used the model GTR + I + G. Bayesian Inference (BI) and Maximum Likelihood (ML) analyzes were performed for the Cytb gene and the COI gene using the evolution models obtained. We ran the BI analysis with MrBayes 3.2 (Ronquist et al., 2012) with the following settings: four chains ran for 10,000,000 generations, with sampling every 1,000 generations and a burn-in of 0.25. We evaluated convergence by the effective sample size (EES) and the potential scale reduction factor (PSRF). For most of the parameters the EES should be ≥200 and for the PSRF most of the values of the parameters should be between 1.0 and 1.2. We conducted the ML analysis with RAxML 8.2.10 (Stamatakis, 2014), using the GTRGAMMA model for all gene matrices, with 10 alternative runs on randomized maximum parsimony starting trees. We obtained nodal supports with the rapid bootstrapping algorithm under the MRE-based Bootstrapping criterion (1,000 replicates). As outgroups we used the genera that make up the tribe Thomasomyini, where Chilomys is included since the study of Smith & Patton (1999). We deposited the new sequences in GenBank, and all sequences used in the analyses are listed in Supplemental S2. Considering the phylogenetic results, we calculated the uncorrected genetic (p-distances) (intraspecific and interspecific) using the alignment (FASTA) of the Cytb gene with the Mega X software (Kumar et al., 2018).

Species delimitation

We to use single-locus method of species delimitation: the Poisson Tree Processes (PTP; Zhang et al., 2013). In the model we used the BI tree of Cytb.

New zoological taxonomic names

The electronic version of this article in Portable Document Format (PDF) will represent a published work according to the International Commission on Zoological Nomenclature (ICZN), and hence the new names contained in the electronic version are effectively published under that Code from the electronic edition alone. This published work and the nomenclatural acts it contains have been registered in ZooBank, the online registration system for the ICZN. The ZooBank LSIDs (Life Science Identifiers) can be resolved and the associated information viewed through any standard web browser by appending the LSID to the prefix http://zoobank.org/. The LSID for this publication is: urn:lsid:zoobank.org:pub:22604A8F-0472-43EB-8D9F-9503C7AE4419. The online version of this work is archived and available from the following digital repositories: PeerJ, PubMed Central and CLOCKSS.

Results

This study was originally envisioned to produce a complete revision of the genus Chilomys following an integrative approach. The Covid19 pandemic hampered the possibility to inspect crucial American collections, in particular those of the FMNH and Smithsonian Institution (Washington, D.C.) containing important samples from Colombia and Venezuela. Under these circumstances, we opted to redesign the scope to be limited to Ecuadorian populations which are currently included in Chilomys instans (see Tirira, 2017).

In the first specimens obtained, we detected noticeable external differences between them, not only in terms of general body size or coloration, but especially in the morphology of manus and pes (e.g., hairiness, distance among pads, patterns of scales). These field observations triggered our interest to conduct an extensive analysis of Cytb sequences of the collected specimens. In addition, the large sample collected in Reserva Drácula (about 50 individuals) allowed us to expand the knowledge of morphological non-geographic variability. Combining the topology of the Cytb marker, the genetic distances, and accounting for ontogenetic and sexual variation, we concluded that C. instans represents a species complex. In the following sections, the main results of the phylogenetic and metric analyses are presented, while the morphological evidence is restricted to taxonomic accounts to avoid redundancy.

Phylogeny

We obtained 21 sequences from the Cytb gene (800 to 1,140 base pairs), while from the COI gene we obtained 14 sequences (560 to 657 base pairs). The genus Chilomys was recovered as monophyletic (Cytb, PP: 1.00/BS: 96; COI, 1.00/100; Fig. 1A) and embedded in a clade with Rhipidomys, and Thomasomys, all recognized members of Thomasomyini. The relationships among these genera differ among the individual genes used. The Cytb gene recovered (Rhipidomys (Thomasomys + Chilomys)) with high supports (PP > 0.90/BS > 70; Fig. 1A), COI recovered (Rhipidomys (Thomasomys + Chilomys)) in some cases the supports were high (0.98/100; Fig. 1B).

Figure 1 Phylogenetic tree based on the mitochondrial gene Cytochrome b (Cytb) and maximum likelihood tree of the mitochondrial Cytochrome Oxidase I (COI).

The dashed line indicates the closest genera: [Rhipidomys + (Thomasomys + Chilomys)]. (A) Bayesian Inference and Maximum Likelihood tree the Cytochrome b; (B) Maximum Likelihood tree of the mitochondrial Cytochrome Oxidase I. In colors the species described within the genus Chilomys: C. georgeledecii sp. nov. (Reserva Drácula), C. instans (Reserva Ecológica El Ángel), C. neisi sp. nov. (El Oro – Zamora Chinchipe), C. percequilloi sp. nov. (Napo – Morona Santiago), C. weksleri sp. nov. (Reserva Integral Otonga). The values above and below the branches represent bootstraps and posterior probability.

Within the Chilomys clade several minor clades were recovered. Cytb topology is resolved in five subclades (Fig. 1A): A group of samples from northern Ecuador, from the Provincia de Carchi, were grouped into two sister clades (1.00/93), one from Reserva Drácula (1.00/98), the other including a sample from Colombia (AF108679) and a group from the Reserva Ecológica El Angel (1.00/96); another group of samples are from the Provincia de Cotopaxi from the Reserva Integral Otonga (1.00/100); and the two remaining clades included samples from the north and south east, two sister clades, one with samples from the Provincias de El Oro and Zamora Chinchipe (1.00/100), and the other with samples from the Provincias de Napo and Morona Santiago (1.00/100). COI recovered four of the five clades found in the phylogenetic tree of Cytb (Fig. 1B): Reserva Ecológica El Angel (1.00/100), Reserva Integral Otonga (1.00/100), Napo and Morona Santiago (0.97/70), and two separate samples, from Zamora Chinchipe and El Oro; for the samples from the Reserva Drácula, no sequences were obtained for COI. The clade of the genus Chilomys presented an intraspecific distance of 6.56% ± 0.49%, while in clades of the phylogenetic tree of Cytb, genetic distance values ranged from 4.88% (Reserva Drácula versus Reserva Ecológica El Angel) to 10.17% (Napo-Morona Santiago vs Colombia AF108679); all pairwise distances are presented in Table 1.

Table 1 Genetic distances.

		1	2	3	4	5	6	
1	Colombia		0.78	1.06	0.70	0.89	0.91	
2	Reserva Ecológica El Ángel-Carchi	6.97		1.01	0.67	0.87	0.85	
3	El Oro-Zamora Chinchipe	9.01	7.75		0.88	0.76	0.88	
4	Reserva Drácula-Carchi	4.96	4.88	6.12		0.79	0.66	
5	Napo-Morona Santiago	10.17	8.72	6.25	7.41		0.75	
6	Reserva Integral Otonga-Cotopaxi	9.35	8.09	7.12	5.85	7.79		
Note:

Matrix of corrected genetic distances (expressed as %, below the diagonal) of Cytochrome b (Cytb) gene sequences among clades of the genus of rodent Chilomys; values above the diagonal are the standard deviation.

Species delimitation

The PTP model (Fig. 2) identified nine putative species (PS): the sample from Colombia and those from Reserva El Angel were identified as different PS1 (1.00) and PS2 (0.98); the Reserva Drácula samples were identified as PS3 (0.99) and PS4 (1.00); the samples from El Oro and Zamora Chinchipe were identified as PS5 (0.84) and PS6 (0.84), respectively; the samples from Napo and Morona Santiago were identified as a single putative species PS8 (0.87), with the exception of the sample QCAZ 8876 which was identified as a different putative species PS7 (1.00); finally, the samples from the Reserva Integral Otonga were identified as PS9 (0.98).

Figure 2 Delimitation of the Poisson Tree Process (PTP) model based on the Cytb maximum likelihood phylogenetic tree for the genus Chilomys.

Lineages (putative species) are identified with blue vertical bars; individuals of the same putative species are denoted in red. The values on the branches represent the posterior probability values (>0.90 values are considered as high support).

Morphometric analysis

The new species Chilomys carapazi is the largest and most distinctive in our sample, and it is evident that most of the variation within the PCA is driven by size along PC1 explaining 68.03% of the variation (Fig. 3A). Among the remaining species there is a large overlap particularly between C. instans and the new species C. percequilloi. The DFA shows that it is possible to differentiate the six species of Chilomys analyzed, although there is some peripheral overlap between C. instans and C. percequilloi, with DF1 explaining 30.54% of the variation (Fig. 3B). The variable breadth of zygomatic plate (BZP) has the strongest effect in both the PCA and DFA, while depth of upper incisor (DI) and breadth of incisive foramina (BIF) have important weights in the PCA, and condyle-incisive length (CIL) and length of nasals (LN) are important for the DFA. The loadings of the first two PCs and the two DFs are presented in Table 2.

Figure 3 Morphometric analyse.

Morphometric analyses of six species of the genus Chilomys. (A) Scatter plot of the principal component analysis (PCA); (B) scatter plot of the discriminant function analysis (DFA).

Table 2 Loadings and percentage of the explained variation of the principal component analysis.

	Character	PC1	PC2	PC3	DF1	DF2	DF3	
1	CIL	0.229	0.042	0.077	0.411	−0.110	0.011	
2	LM	0.068	0.071	0.335	0.277	−0.175	−0.056	
3	LR	0.199	0.033	0.178	0.352	−0.068	−0.109	
4	BR	0.161	−0.126	0.084	0.344	−0.065	0.266	
5	LN	0.294	0.199	0.231	0.547	0.183	0.001	
6	BB	0.065	0.084	0.158	0.356	0.038	−0.152	
7	BM1	0.130	−0.029	0.380	0.258	−0.058	0.027	
8	LIF	0.246	−0.227	0.112	0.266	−0.175	0.000	
9	BIF	0.110	−0.255	0.328	0.263	−0.453	0.319	
10	BPB	0.060	−0.251	−0.054	0.074	−0.244	0.236	
11	BZP	0.473	0.337	0.000	0.520	0.058	0.116	
12	LIB	0.085	−0.046	0.060	0.329	−0.567	−0.032	
13	ZB	0.151	−0.008	0.024	0.345	−0.104	0.243	
14	BM	0.064	−0.477	0.192	0.060	0.003	−0.007	
15	DI	0.434	0.007	−0.397	0.283	−0.090	0.058	
16	BIT	0.396	−0.478	−0.336	0.345	−0.056	0.031	
17	LLM	0.082	0.048	0.407	0.374	−0.090	−0.105	
18	LMN	0.194	0.032	0.020	0.357	−0.131	0.078	
19	DR	0.217	0.416	−0.142	0.309	0.103	0.372	
	% Variation	68.03	5.45	4.87	30.54	23.69	20.48	
Note:

Loadings and percentage of the explained variation of the Principal Component Analysis (first three principal components) and of the Discriminant Function Analysis (first three discriminant functions) performed on five species of the genus of rodent Chilomys. Acronyms of variables are explained in the main text (Materials and Methods section).

Systematic accounts

Family Cricetidae Fischer, 1817

Subfamily Sigmodontinae Wagner, 1843

Tribe Thomasomyini Steadman and Ray, 1982

Genus Chilomys Thomas, 1897

Chilomys carapazi sp. nov. Brito and Pardiñas

urn:lsid:zoobank.org:act:A12AF0E7-4465-4A9F-99B0-7E09DBDD5BBA

Carapaz’s Forest Mouse, Ratón del bosque de Carapaz (in Spanish)

Holotype: MECN 5291 (field number JBM [Jorge Brito Molina] 1453), an adult male captured 27 September, 2016, by J. Brito, J, Robayo, L, Recalde, T, Recalde and C. Reyes, preserved as a cleaned skull and the rest of the body in ethanol, and muscle and liver biopsies in 95% ethanol.

Type locality: Ecuador, Provincia de Carchi, Reserva Drácula, Gualpi Km 18 (0.849796°, −78.234767°, WGS84 coordinates taken by GPS at the site of collection; elevation 2,350 m).

Etymology: Named in honor of Richard Carapaz Montenegro, an Ecuadorian professional cyclist born in the Provincia de Carchi. The species epithet is formed from the surname “Carapaz,” taken as a noun in the genitive case, adding the Latin suffix “i” (ICZN 31.1.2).

Diagnosis: A species of Chilomys which can be identified by the following combination of characters: Head and body length ~95 mm; dorsal surface of foot covered with round scales and without interspaces; long nasal (~8.5 mm); long diastema (~8.2 mm); M2 with broad hypoflexus (similar in width to mesoflexus); m1 without anteromedian flexid.

Morphological description of the holotype: Large body size for the genus (head and body length combined 95 mm). Brown color (color 277) dorsal fur (Fig. 4); short hairs (medium length on back = 9 mm) with medium neutral gray (color 298) base and ground cinnamon (color 270) tips. Smoke gray (color 267) ventral coat, with hairs (medium length = 7 mm) with dark neutral gray (color 299) base and smoke gray (color 266) tips. Olive-brown (color 278) periocular ring. Postauricular patch absent. Mystacial vibrissae long, thick at base and thin towards tip, exceeding shoulder when tilted backwards; superciliary vibrissae 1 present, genal vibrissae 1 present (sensu Pacheco, 2003). Ears (11 mm from notch to margin) externally covered by short smoke gray (color 266) hairs, and with pale buff (color 1) inner surface and pale neutral gray (color 296) margin.

Figure 4 Chilomys carapazi.

External aspect of Chilomys carapazi sp. nov., in its natural habitat (painted by Glenda Pozo).

Narrow and ground cinnamon drab (color 259) metatarsal patch, which extends to the base of the phalanges; dorsal surface of the foot with round scales and without interspaces (Fig. 5A). Plantar surface with six pads, including four interdigitals of similar size, thenar and hypothenar pads large and with ample interspace; sole between pads smooth (Fig. 5B). Short digit I reaches base of digit II; digit II slightly smaller than digit III and digit III same size as digit IV; short digit V (apparently somewhat opposable) reaches middle of digit IV. Long tail (95 mm; 134% of HB), unicolor fawn (color 258) except for apex, which is white (up to 10 mm). Tail with 16 rows of scales per cm on axis; rectangular scales with three hairs each, which extend over 1–1.5 rows of scales; naked-looking tail except for tip, where it presents a small brush of up to 5 mm. Prominent anus.

Figure 5 Morphology comparisons.

Morphology of the dorsal (upper row) and ventral (lower row) surface of the right hind foot in species of Chilomys. (A, B) Chilomys carapazi sp. nov. (MECN 5291, holotype; Reserva Drácula, Carchi, Ecuador); (C, D) C. georgeledecii sp. nov. (MECN 6337, paratype; Reserva Drácula, Carchi, Ecuador); (E, F) C. neisi sp. nov. (MECN 6187, holotype; Ashigsho, Chilla, El Oro, Ecuador); (G, H) C. percequilloi sp. nov. (MECN 6362, paratype; Parque Nacional Llanganates, Tungurahua, Ecuador); (I, J) C. weksleri sp. nov. (MECN 6365, holotype; Reserva Geobotánica Pululahua, Pichincha, Ecuador). Approximately scaled to the same length. Photographs by J. Brito.

Cranium large for the genus (26.35 mm of CIL). Short and narrow rostrum, with nasal bones that do not extend to incisors; poorly developed gnathic process. Posterior margin of nasal bone not surpassing plane of lacrimal bone. Shallow zygomatic notch. Large and rounded lacrimal bones. Wide interorbital region with smooth outer edges, without exposing alveolar maxillary processes in dorsal view (Fig. 6A). Supraorbital region with diverging posterior borders. Frontoparietal suture U-shaped. Broad, rounded and not inflated braincase, concave at outer edges. Broad zygomatic plate, comparatively longer than length of M1, leaning forward and with posterior edge not reaching maxillary row. Zygomatic arches sturdy with jugals spanning a large segment of each mid-arch. Small supraorbital foramen with posterior border in line with M3 (Fig. 7B). Alisphenoid strut present but narrow. Carotid circulatory pattern type 3 (sensu Voss, 1988); carotid canal large, stapedial foramen small, without alisphenoid squamous groove and with sphenofrontal foramen. Subsquamosal fenestra four times larger than postglenoid foramen; hamular process of squamosal thin and long, and distally applied on mastoid capsule. Slightly triangular tegmen tympanic, superimposed with suspensory process of squamous. Lateral expressions of parietals present; bullae small; pars flaccida of tympanic membrane present; orbicular apophysis of malleus well-developed. Paraoccipital process small. Hill foramen small; short and wide incisive foramina with curved edges, not reaching plane defined by anterior faces of M1. Premaxillary capsule narrow, parallel-sided and narrow at rear ends; maxillary septum of incisive foramen slim and long. Long and wide palate (sensu Hershkovitz, 1962). Posterolateral palatal pit small. Wide mesopterygoid fossa, with by a medium palatal process present. Inconspicuous sphenopalatine vacuities covered by roof of palate. Basisphenoid wide. Large foramen ovale, similar in size to transverse canal. Middle lacerate foramen narrow. Auditory bullae small and uninflated with large and narrow eustachian tube (Fig. 8B).

Figure 6 Chilomys carapazi sp. nov.

Chilomys carapazi sp. nov. (Reserva Drácula, Carchi, Ecuador): cranium in (A) dorsal, (B) ventral, and (C) lateral views, and mandible in (D) labial view (MECN 5291 holotype). Photographs by J. Brito.

Figure 7 Morphological comparisons.

Comparison of the left anterior portion of the cranium, viewed from the side, in several species of Chilomys: (A) C. instans (NHMUK 1895.10.14.1, holotype); (B) C. carapazi sp. nov. (MECN 5291, holotype); (C) C. georgeledecii sp. nov. (MECN 6024, holotype); (D) C. neisi sp. nov. (MECN 6187, holotype); (E) C. percequilloi sp. nov. (MECN 5854, holotype); and (F) C. weksleri sp. nov. (MECN 6365, holotype). Thomas’ angles according to incisive and basal planes are indicated as well as the extension of the molar series. Abbreviations: nc, nasolacrimal capsule; m, masseteric scar; nlf, nasolacrimal fissure; sf, supraorbital foramen; sm, supramaxillary foramen, zp, zygomatic plate. Three-dimensional reconstruction by C. Koch and J. Brito.

Figure 8 Morphological comparisons.

Comparison of right auditory capsule in ventral view in several species of Chilomys: (A) C. instans (NHMUK 1895.10.14.1, holotype); (B) C. carapazi sp. nov. (MECN 5291, holotype); (C) C. georgeledecii sp. nov. (MECN 6024, holotype); (D) C. neisi sp. nov. (MECN 3723, paratype); (E) C. percequilloi sp. nov. (MECN 5854, holotype); and (F) C. weksleri sp. nov. (MECN 6365, holotype). Abbreviations: bet, bony eustachian tube; cc, carotid canal; e, ectotympanic; mlf, middle lacerate foramen; pt, petrosal; sft, stapedial foramen. Three-dimensional reconstruction by C. Koch and J. Brito.

Dentary with short and wide coronoid process (not extending beyond upper edge of condylar process); short and thin mental foramen.

Proodont upper incisors (Thomas angle ~95°; Fig. 7B) with orange and smooth front enamel; lower incisors with sharp tip; crested and pentalophodont molars (sensu Hershkovitz, 1962), with noticeably thick enamel. Maxillary molar rows converging slightly backwards; main cusps opposite (Fig. 9A) and sloping backwards when viewed from side. M1 rectangular in outline; without anteromedian flexus; deep paraflexus; short and wide anteroloph; short and wide mesoloph; reduced posteroloph. M2 squared in outline; mesoloph showing same condition as in M1; broader hypoflexus (similar in width to mesoflexus); internal fosseta larger than fosseta of M1. M3 less than half the size of M2; M3 rounded in outline with conspicuous anteroloph; central fosseta small. Lower molars with opposite main cusps (Fig. 9B) and sloping forwards when viewed from side. First lower molar (m1) without anteromedian flexus; large anterolabial cingulum; short mesolophid; mesolophid of m2 showing same condition as in m1; noticeable anterolabial cingulum; hypoflexid of m3 long and wide.

Figure 9 Morphological comparisons.

Comparison of upper (A, C, E, G, I) and lower (B, D, F, H, J) right molar series in occlusal view among species of Chilomys: (A, B) C. carapazi sp. nov. (MECN 5291, holotype); (C, D) C. georgeledecii sp. nov. (MECN 6024, holotype); (E, F) C. neisi sp. nov. (MECN 6187, holotype); (G, H) C. percequilloi sp. nov. (MECN 6338, paratype); and (I, J) C. weksleri sp. nov. (MECN 6363, paratype). Abbreviations: al, anterolabial cingulum; am, anterior mure; af, anteromedian flexus/id; h, hypoflexid; m, mesoloph/id; mm, median murid; p, protoflexus. Photographs by J. Brito.

Comparisons

Chilomys carapazi sp. nov., is the largest species recognized for the genus (Fig. 3; Supplemental S3). As it occurs in sympatry with C. georgeledecii sp. nov., at Reserva Drácula it could be confused with this species. Nevertheless, beside metric characteristics (see Table 3) it differs from C. georgeledecii (states in parenthesis) by the following traits: Thomas angle ~95° (Thomas angle ~102°); M1 without anteromedian flexus (with anteromedian flexus); M2 with broader hypoflexus, similar in width to mesoflexus (with narrowed hypoflexus, distinctly narrower than mesoflexus); m1 without anteromedian flexus (m1 with anteromedian flexus). A detailed comparison with all species of Chilomys is presented in Table 3.

Table 3 Morphological comparisons.

C. carapazi	C. georgeledecii	C. percequilloi	C. neisi	C. weksleri	C. instans	C. fumeus	
Head and body length ~95 mm	Head and body length range 83–90 mm	Head and body length range 76–90 mm	Head and body length range 95–100 mm	Head and body length range 74–85 mm	Head and body length range 78–98 mm	Head and body length range 86–90 mm	
Tail ~134% head-body length	Tail ~144, 44–177.78% head-body length	Tail ~137–155% head-body length	Tail ~134–136% head-body length	Tail ~143–153 head-body length	Tail ~112–146% head-body length	Tail ~134–139% head-body length	
Tail with 16 rows of scales per cm on the axis	Tail with 16–18 rows of scales per cm on	Tail with 18–20 rows of scales per cm on the axis	Tail with 15–16 rows of scales per cm on the axis	Tail with 16 rows of scales per cm on the axis	Tail with 17–19 rows of scales per cm on the axis	Tail with 16 rows of scales per cm on the axis	
Dorsal surface of the foot with round scales and without interspaces	Dorsal surface of the foot with round scales and large interspaces	Dorsal surface of the foot with round scales and small interspaces	Dorsal surface of the foot with round scales and small interspaces	Dorsal surface of the foot with round scales and small interspaces	–	Dorsal surface of the foot with round scales and small interspaces	
Large thenar and hypothenar pads, ample interspace	Large thenar and hypothenar pads, small interspace	Hypothenar smaller than thenar pad	Small thenar and hypothenar pads, ample interspace	Large thenar and hypothenar pads, small interspace	Large thenar and hypothenar pads, small interspace	–	
Interspace between pads smooth	Interspace between pads smooth	Interspace between pads scaly	Interspace between pads smooth	Interspace between pads smooth	Interspace between pads smooth	Interspace between pads smooth	
Nasals long 8.55 mm	Nasals short 6.1–7.5 mm	Nasal long 7.2–8.8 mm	Nasals long 8.4–8.8 mm	Nasals long 7.2–7.7 mm	Nasals long 7.4–8.76 mm	Nasals short 6.41–7.39 mm	
Broad zygomatic plates 2.60 mm	Narrowed zygomatic plates 1.66–2.09 mm	Broad zygomatic plates 2.03–2.3 mm	Broad zygomatic plates 2.1–2.4 mm	Narrowed zygomatic plates 1.8–2.1 mm	Broad zygomatic plates 2.05–2.41 mm	Narrowed zygomatic plates 1.34–1.96 mm	
Diastema long 8.23 mm	Diastema long 6.5–7.4 mm	Diastema long 7–7.58 mm	Diastema long 7.33 mm	Diastema long 6.8–7.39 mm	Diastema long 7.0– 7.91 mm	Diastema short 6.14– 6.96 mm	
Thomas angle 95°	Thomas angle 102°	Thomas angle 92°	Thomas angle 102°	Thomas angle 92°	Thomas angle 100°	Thomas angle 94°	
M1 without flexus anteromedian	M1 with flexus anteromedian	M1 with flexus anteromedian	M1 without flexus anteromedian	M1 with flexus anteromedian	M1 with flexus anteromedian	M1 with flexus anteromedian	
M1–M2 present Mesoloph	M1–M2 present Mesoloph	M1–M2 present Mesoloph	M1–M2 indistinct Mesoloph	M1–M2 present Mesoloph	M1–M2 present Mesoloph	M1–M2 present Mesoloph	
M2 with broader hypoflexus (similar in width to mesoflexus)	M2 with narrowed hypoflexus (distinctly narrower than mesoflexus)	M2 with broader hypoflexus (similar in width to mesoflexus)	M2 with narrowed hypoflexus (distinctly narrower than mesoflexus)	M2 with broader hypoflexus (similar in width to mesoflexus)	M2 with broader hypoflexus (similar in width to mesoflexus)	M2 with narrowed hypoflexus (distinctly narrower than mesoflexus)	
Maxillary toothrow large 3.4 mm	Maxillary toothrow large <3.2 mm	Maxillary toothrow large <3.4 mm	Maxillary toothrow large <3.2 mm	Maxillary toothrow large <3.3 mm	Maxillary toothrow large >3.1 mm	Maxillary toothrow short <3.1 mm	
m1 without flexus anteromedian	m1 with flexus anteromedian	m1 with flexus anteromedian	m1 without flexus anteromedian	m1 with flexus anteromedian	m1 without flexus anteromedian	m1 without flexus anteromedian	
–	Hemal arches present	Hemal arches present	Hemal arches absent	Hemal arches present	Hemal arches Absent	Hemal arches present	
This study	This study	This study	This study	This study	Pacheco, 2015a; this study	Osgood, 1912; Pacheco, 2015a; this study	
Note:

Morphological comparisons of selected traits among species of the genus of rodent Chilomys.

Distribution: Known only from the type locality at Reserva Drácula (Carchi, Ecuador), on the western flank of the Andes (Fig. 10), at an elevation of 2,350 m. The climate at this locality has an average annual temperature of 15.5 °C and a precipitation of 1,520 mm per year. The climate is relatively stable during the first months of the year and between July and October the differences between minimum and maximum temperatures increase, with the lowest temperatures in August (9.3 °C) and the highest in September (21.8 °C). The highest precipitation occurs in October with an average of 190 mm per month, the lowest in August with 46 mm per month (Hijmans et al., 2005).

Figure 10 Chilomys in Ecuador.

Localities for the species of Chilomys recognized in Ecuador. Symbols with a black dot in the center represent type localities.

Natural history: The type locality is located in the headwaters of the Gualpi River in the lower montane ecosystem (Cerón et al., 1999). The local expression of the montane cloud forest is characterized by a tree canopy that reaches 30 m high. The understory is luxurious and mostly composed of species belonging to Araceae, Melastomataceae, Cyclanthaceae, Bromeliaceae, and ferns. From the same pit falls where Chilomys carapazi sp. nov., was obtained, we also collected the sigmodontines C. georgeledecii, Pattonimus ecominga, Melanomys caliginosus, Microryzomys minutus, Nephelomys cf. pectoralis, and Thomasomys bombycinus, the heteromyid Heteromys australis, the marsupials Caenolestes convelatus, Mamosops caucae, and the soricid Cryptotis equatoris.

Chilomys georgeledecii sp. nov. Brito, Tinoco, García, Koch and Pardiñas

urn:lsid:zoobank.org:act:BDEFF98C-5ED9-4DC7-8EC9-6ADE8BB297C1

Ledeci Forest Mouse, Ratón del bosque de Ledeci (in Spanish)

Holotype: MECN 6024 (field number JBM 1955), an adult male captured 8 November, 2018, by J. Brito, J, Curay and R, Vargas, preserved as dry skin, skull, postcranial skeleton and muscle and liver biopsies in 95% ethanol.

Paratypes: MECN 4732, MECN 4751, and MECN 4752, adult males, and MECN 4761, adult female, all preserved as cleaned skulls and carcasses in ethanol, collected in Provincia de Carchi, Reserva Drácula, Cerro Oscuro (0.917274°, −78.187079°, 1,550 m) by J. Brito, J. Robayo, L. Recalde, T. Recalde and C. Reyes on 7 July, 2015. MECN 4983, MECN 4992, MECN 4993, MECN 4994, MECN 4995, MECN 4996, and MECN 4997, adult males, MECN 4925, MECN 4955, and MECN 4956, adult females, all preserved as dry skins and cleaned skulls, collected in Gualpi Km 14 (0.882408°, −78.223235°, 1,970 m) by J. Brito, J. Robayo, L. Recalde, T. Recalde and C. Reyes on 5 June, 2016. MECN 4968, MECN 4971 and MECN 5381, adult males preserved as dry skins and cleaned skulls, collected in Gualpi Km 18 (0.849796°, −78.234767°, 2,350 m) by J. Brito, J. Robayo, L. Recalde, T. Recalde and C. Reyes on 2 June, 2016. MECN 5301, MECN 5302, MECN 5303, adult males, MECN 5299, MECN 5300 adult females, all preserved as cleaned skulls and carcasses in ethanol, collected in Gualpi Km 18 (0.849796°, −78.234767°, 2,350 m) by J. Robayo, J. Brito and H. Yela on 27 September, 2016. MECN 5921, MECN 5925, and MECN 6205, adult males, MECN 5923 and MECN 5926, adult females, preserved as dry skins and cleaned skulls, collected in Guapilal (0.891944°, −78.20308°, 1,700 m) by J. Curay, R. Vargas and C. Bravo on 14 April, 2019. MECN 6323, MECN 6327, and MECN 6337, adult males, MECN 6303, an adult female, preserved as dry skins and cleaned skulls, collected in Bosque La Esperanza (0.929830°, −78.244860°, 1,912 m) by J. Brito, J. Castro, Z. Villacis and J. Guaya on 28 March, 2021.

Type locality: Ecuador, Provincia de Carchi, Reserva Drácula, Peñas Blancas-Pailón (−0.98259°, −78.22204°, WGS84 coordinates taken by GPS at the site of collection; elevation 1,502 m).

Etymology: Named in honor of Czech and US international conservationist George Campos Ledeci, who has worked to promote more environmentally friendly infrastructure development projects in Ecuador and other countries. The species epithet is formed from the surname “Ledeci,” taken as a noun in the genitive case, adding the Latin suffix “i” (ICZN 31.1.2).

Diagnosis: A species of Chilomys which can be identified by the following combination of characters: Head and body length ~83–90 mm; tail longer than head and body length combined (~144.4–177.7%); dorsal surface of foot with round scales and large interspaces; zygomatic plate slightly tilted backwards; M2 with narrow hypoflexus (distinctly narrower than mesoflexus); m1 with anteromedian flexus.

Morphological description of the holotype and variation: Small body size for the genus (head and body length combined range between 76 and 90 mm). Medium neutral gray (color 298) dorsal fur; short hairs (medium length on back = 5.5 mm). Pale neutral gray (color 296) venter coat, with hairs (medium length = 6.5 mm) with dark natural neutral gray (color 299) base. Jet black (color 300) periocular ring (Fig. 11). Postauricular patch absent. Mystacial vibrissae short, thick at base and thin towards tip, slightly exceeding ears when are tilted backwards; superciliary vibrissae 1 present, genal vibrissae 1 present. Ears (11–16 mm from notch to margin) externally covered by short smoke gray (color 266) hairs, and with dark neutral gray (color 299) inner surface and light neutral gray (color 296) margin (Fig. 11).

Figure 11 External aspect of Chilomys georgeledecii sp. nov.

External aspect of Chilomys georgeledecii sp. nov. (MECN 6024, holotype), an adult male from Reserva Drácula, Carchi, Ecuador. Photograph by J. Brito.

Metatarsal patch with whitish hairs, giving a naked; dorsal surface of foot with round scales and large interspaces. Plantar surface with 6 pads, including 4 interdigitals of similar size, thenar and hypothenar pads large and with small interspace; sole between pads is smooth (Fig. 5D). Short digit I reaches base of digit II; digit II slightly smaller than digit III and digit III slightly smaller than digit IV; short digit V reaches middle of digit IV. Long tail (120–140 mm; ~144.44–177.78% of HB), unicolor fawn (color 258) except for apex, which is white (up to 12–20 mm). Tail with 16–18 rows of scales per cm on axis; square scales with three hairs each, which extend over 1.5 rows of scales; naked-looking tail except for tip, where it presents a small brush of up to 4 mm.

Cranium small for the genus (20.8–23.3 mm of CIL). Short and narrow rostrum, with nasal bones that extend to incisors; poorly developed gnathic process. Posterior margin of nasal bone does not exceed plane of lacrimal bone. Shallow zygomatic notch. Small and triangular outline of lacrimal bones, almost entirely welded to maxillae. Wide interorbital region with smooth outer edges, without exposing alveolar maxillary processes in dorsal view (Fig. 12A). Supraorbital region with diverging posterior borders. Frontoparietal suture U-shaped. Broad rounded and inflated braincase, concave at outer edges. Developed ethmoturbinals (Fig. 13F). Narrow zygomatic plate, comparatively same length as M1, leaning forward and with posterior edge not reaching maxillary row. Zygomatic arches thin with jugals spanning a large segment of each mid-arch. Large supraorbital foramen with posterior border in line with M2 (Fig. 7C). Alisphenoid strut wide and robust. Carotid circulatory pattern type 3; carotid canal large, stapedial foramen very small, without alisphenoid squamous groove and without sphenofrontal foramen. Subsquamous fenestra three times smaller than postglenoid foramen (Fig. 12C); hamular process of squamosal thin and long, and distally applied on mastoid capsule. Triangular tegmen tympanic, superimposed with suspensory process of squamous. Lateral expressions of parietals present; bullae small; pars flaccida of tympanic membrane present, large; orbicular apophysis of malleus well-developed. Paraoccipital process small. Hill foramen small (Fig. 14C); short and narrow incisive foramina with curved edges without reaching plane defined by anterior faces of M1; a pair of ridges on either side of palatine foramina and in front of M1. Premaxillary capsule widened, parallel-sided and narrow at rear ends; maxillary septum of incisive foramen slim and long. Long and wide palate. Posterolateral palatal pits small. Wide mesopterygoid fossa, with by a medium palatal process present. Inconspicuous sphenopalatine vacuities covered by roof of palate. Basisphenoid narrow. Small foramen ovale, but larger than transverse canal. Middle lacerate foramen narrow. Auditory bullae small and uninflated with short and wide eustachian tubes (Fig. 8C).

Figure 12 Chilomys georgeledecii sp. nov.

(Reserva Drácula, Carchi, Ecuador): cranium in (A) dorsal, (B) ventral, and (C) lateral views, and mandible in (D) labial view (MECN 6024, holotype). Three-dimensional reconstruction by C. Koch and J. Brito.

Figure 13 Morphological comparisons.

Comparison of selected regions of the cranium in several species of Chilomys, including the basicraneal region (upper row; roofing bones of braincase removed) in dorsal view and the cross section at the frontal sinuses plane (lower row): (A, E) C. instans (NHMUK 1895.10.14.1, holotype); (B, F) C. georgeledecii sp. nov. (MECN 6024, holotype); (C, G) C. neisi sp. nov. (MECN 3723, paratype), and (D, H) C. percequilloi sp. nov. (MECN 5854, holotype). Abbreviations: bo, basioccipital; bs, basisphenoid; cc, carotid canal; etl-lll, ethmoturbinals; fo, foramen ovale; ft1-2, frontoturbinals; it, interturbinal; lc, lamina cribosa; ls, lamina semicircularis; pet, petrosal; ps, presphenoid; sacg, groove for secondary arterial connection; sact, tunnel-like medial entrance to alisphenoid canal for secondary arterial connection. Three-dimensional reconstruction by J. Brito.

Figure 14 Morphological comparisons.

Comparison of diastemal palate in several species of Chilomys: (A) C. instans (NHMUK 1895.10.14.1, holotype); (B) C. carapazi sp. nov. (MECN 5291, holotype); (C) C. georgeledecii sp. nov. (MECN 6024, holotype); (D) C. neisi sp. nov. (MECN 3723, paratype); (E) C. percequilloi sp. nov. (MECN 5854, holotype); and (F) C. weksleri sp. nov. (MECN 6365, holotype). Arrows in (D) point to masseteric ridges; abbreviations: hf, Hill foramen; m, masseteric scar. Three-dimensional reconstruction by C. Koch and J. Brito.

Dentary with short and narrow coronoid process (extends beyond upper edge of condylar process); short and thin mental foramen.

Proodont upper incisors (Thomas angle ~102°; Fig. 7C) with orange and smooth front enamel; crested and pentalophodont molars, with noticeably thick enamel. Maxillary molar rows parallel; main cusps opposite (Fig. 9C) and sloping backwards when viewed from side. M1 rectangular in outline with anteromedian flexus; conspicuous anteroloph; long and wide mesoloph; short posteroloph; internal closure of mesoflexus ends in a fosseta. M2 squared in outline; mesoloph showing same condition as in M1; narrow hypoflexus (distinctly narrower than mesoflexus); internal fosseta larger than fosseta of M1. M3 less than half the size of M2; M3 rounded in outline with conspicuous anteroloph; long paraflexus; central fosseta large, but smaller than in M2 (Fig. 9C). Lower molars with main cusps opposite (Fig. 9D) and sloping forwards when viewed from side. First lower molar (m1) with anteromedian flexus; small anterolabial cingulum; thin and long mesolophid. Mesolophid of m2 showing same condition as in m1; conspicuous cingulum m2. Hypoflexid of m3 long and wide; hypoflexid well-developed and deep in m1–m3 (Fig. 9D).

Tuberculum of first rib articulates with transverse processes of seventh cervical. First and second thoracic vertebrae have differentially elongated neural spine. Vertebral column composed of 19 thoracicolumbar, 16th with moderately developed anapophyses and 17th with little developed anapophyses, 4 sacrals (fused), and 37–43 caudal vertebrae; hemal arches in third and fourth caudal vertebra; 12 ribs.

Comparisons: Chilomys georgeledecii sp. nov., is one of the smallest species of Chilomys that inhabits Ecuador (Fig. 3; Table 4). It occurs in sympatry with C. carapazi sp. nov., at Reserva Drácula and can be confused with this species. Nevertheless, beside metric characteristics (see Table 4) it differs from C. caparazi (states in parenthesis) by the following traits: Thomas angle ~102°; M1 with anteromedian flexus; M2 with narrow hypoflexus, distinctly narrower than mesoflexus (broader and similar in width to mesoflexus in C. carapazi); m1 with anteromedian flexus.

Table 4 Comparative statistic.

	C. carapazi sp. nov.	C. georgeledecii sp. nov.	C. percequilloi sp. nov.	C. neisi sp. nov.	C. weksleri sp. nov.	C. instans	
Holotype	Holotype	Paratypes	Holotype	Paratypes	Holotype	Paratype	Holotype	Paratypes	Holotype	
MECN 5291	MECN 6024		SD	max	min	N	MECN 5854		SD	max	min	N	MECN 6187		MECN 6365		SD	max	min	N	NHMUK 1895.10.14.1	
HBL	95	80	78	8.37	90	63	24	95	86	6.94	95	70	18	95	100	75	75.8	8.46	85	65	5	99	
LT	128	122	119	9.50	140	100	24	133	114	9.53	132	96	18	128	136	103	110	7.68	121	105	5	130	
HF	24	24	23.4	1.44	25	19	24	26	23.1	2.31	28	20	18	25	27	21	23	2.45	26	20	5	22.70	
E	11	14	13.4	1.53	16	10	24	16	15.1	1.96	17	8	18	16	15	14	14.80	0.96	16	14	5	14.10	
CIL	26.35	22.23	21.7	1.11	23.3	19.3	24	24.24	23.4	1.07	24.6	20.8	18	24.05	24.70	21.88	22.10	1.41	23.55	20	5	24.16	
CBL	26.31	22.31	21.9	1.01	23.6	19.8	24	24.66	23.6	1.09	25	21	18	24.24	24.90	22.15	22.30	1.36	23.62	20.30	5	22.93	
LD	8.23	6.93	6.72	0.41	7.40	5.86	24	7.58	7.20	0.36	7.70	6.39	18	7.33	7.30	6.81	6.88	0.48	7.39	6.12	5	7.15	
LM	3.43	3.19	3.11	0.08	3.30	2.99	24	3.33	3.24	0.10	3.40	3	18	3.24	3.20	3.11	3.09	0.19	3.30	2.80	5	3.47	
LR	7.87	7.09	6.84	0.30	7.26	6.01	24	7.92	7.34	0.37	7.80	6.60	18	7.04	7.90	6.73	7.01	0.44	7.61	6.53	5	6.60	
BR	5.18	4.3	4.28	0.21	4.80	3.90	24	4.93	4.52	0.16	4.79	4.10	18	4.58	4.90	4.03	4.29	0.19	4.54	4.04	5	4.31	
LN	8.55	6.55	6.87	0.43	7.55	6.04	24	8.41	7.88	0.55	8.80	6.30	18	8.41	8.80	7.35	7.31	0.35	7.76	6.99	5	7.41	
BB	11.71	11.57	11.2	0.26	11.7	10.7	24	12.14	11.7	0.24	12.15	11.20	18	11.87	11.80	11.08	11.3	0.44	12.10	11	5	11.26	
BM1	1.12	1.07	0.98	0.04	1.07	0.90	24	1.20	1.05	0.06	1.14	1	18	1.19	1	0.93	0.98	0.07	1.07	0.90	5	1.11	
LIF	4.03	3.73	3.37	0.27	3.70	2.81	24	3.73	3.53	0.27	3.90	2.90	18	3.96	3.90	3.31	3.30	0.14	3.53	3.18	5	3.69	
BIF	1.77	1.32	1.35	0.08	1.50	1.21	24	1.37	1.37	0.06	1.50	1.29	18	1.32	1.50	1.29	1.23	0.09	1.33	1.09	5	1.64	
BPB	2.75	2.66	2.46	0.09	2.66	2.30	24	2.68	2.43	0.09	2.60	2.20	18	2.55	2.70	2.22	2.43	0.16	2.70	2.30	5	2.46	
BZP	2.60	1.69	1.66	0.17	2.09	1.31	24	2.10	2.03	0.14	2.30	1.70	18	2.12	2.40	1.66	1.86	0.19	2.10	1.57	5	1.72	
LIB	5.29	4.76	4.69	0.10	4.89	4.56	24	4.78	4.79	0.13	5.10	4.60	18	4.80	4.80	4.37	4.53	0.18	4.70	4.23	5	4.70	
ZB	14.98	13.17	12.6	0.44	13.23	11.8	24	13.9	13.1	0.39	13.6	12.2	18	13.4	14.10	12.34	12.8	0.63	13.35	11.80	5	13.22	
BM	1.34	1.35	1.33	0.09	1.50	1.18	24	1.54	1.35	0.11	1.54	1.14	18	1.38	1.40	1.38	1.29	0.08	1.40	1.20	5	1.71	
DI	1.65	1.16	1.17	0.12	1.34	0.90	24	1.44	1.30	0.13	1.50	1	18	1.35	1.50	1.07	1.24	0.19	1.40	0.94	5	1.23	
BIT	1.54	1.17	1.19	0.11	1.45	0.95	24	1.52	1.34	0.14	1.50	1	18	1.65	1.60	1.11	1.21	0.16	1.40	0.96	5	1.31	
LLM	3.54	3.29	3.24	0.07	3.40	3.12	24	3.54	3.43	0.14	3.70	3.10	18	3.43	3.40	3.01	3.22	0.10	3.35	3.10	5	3.36	
LMN	15.89	13.77	13.4	0.68	14.4	11.70	24	14.8	14.1	0.57	15	12.40	18	14.55	15.4	13.28	13.50	0.65	14.07	12.70	5	13.72	
DR	3.30	2.43	2.43	0.12	2.70	2.28	24	2.70	2.63	0.16	3	2.40	18	2.83	2.90	2.60	2.74	0.28	3.03	2.44	5	2.32	
W	–	18.50	19.07	4.21	24	13	7	23	18	2.54	23	15	11	16	27	–	18.50	4.95	22	15	2	–	
Note:

Univariate statistics (, mean; SD, standard deviation; max, maximum; min, minimum; N, number of specimens) and external and craniodental measurements (in mm), and weight in grams for each species of the genus Chilomys; measured specimens are listed in Appendix 1; acronyms are explained in the main text.

Another species that inhabits the western flank of Ecuador (Fig. 10) and is similar in size to C. georgeledecii is Chilomys weksleri (named below, see Table 4); C. georgeledecii differs from C. weksleri sp. nov., (states in parentheses) by the following traits: zygomatic plate comparatively same length as M1 and slightly tilted backwards (comparatively wider than M1 and leaning forwards); M2 with narrow hypoflexus, but distinctly narrower than mesoflexus (broader hypoflexus, similar in width to mesoflexus). Further comparison with all recognized species of Chilomys is provided in Table 3.

Distribution: Known from several neighbouring collecting sites in Reserva Drácula (Carchi, Ecuador), on the western flank of the Andes (Fig. 10), at elevations ranging from 1,502 to 2,350 m. The climate in the recorded localities has an annual mean temperature of 18 °C and precipitation of 1,720 mm per year. The greatest differences between minimum and maximum temperatures occur between June and October, with the lowest monthly average temperature in August (9.3 °C) and the highest in September (25.2 °C). The highest precipitation occurs in April with an average of 230 mm per month and the lowest in July and August with 30 mm per month each (Hijmans et al., 2005).

Natural history: Reserva Drácula belongs to the subtropical and lower montane ecosystem (Cerón et al., 1999). The local expression of the cloud montane forest is characterized by a tree canopy that reaches 30 m high. The understory is luxurious and mostly composed of species belonging to Araceae, Melastomataceae, Cyclanthaceae, Bromeliaceae, and ferns. Stomachs from six specimens were dissected to inspect content (Supplemental S4). Sampled C. georgeledecii sp. nov., were insectivorous, preying primarily on fly larva. Identifiable prey items were 50% Diptera, 28.5% Coleoptera, 7.1% Hymenoptera, 7.1% Blattodea, and 7.1% Annelida. From the same pit falls where C. georgeledecii sp. nov., was obtained, we also collected the sigmodontines Chilomys carapazi sp. nov., Pattonimus ecominga, Melanomys caliginosus, Microryzomys minutus, Nephelomys cf. pectoralis, Oecomys sp., Rhipidomys latimanus, Tanyuromys thomasleei, Sigmodontomys alfari, and Thomasomys bombycinus, the heteromyid Heteromys australis, the marsupials Caenolestes convelatus, Mamosops caucae, and Marmosa isthmica, and the soricid Cryptotis equatoris.

Chilomys neisi sp. nov. Brito, Tinoco, García, Koch, and Pardiñas

urn:lsid:zoobank.org:act:F31C845C-DED1-4579-992D-9602FF14ADA6

Neisi Forest Mouse, Ratón del bosque de Neisi (in Spanish)

Holotype: MECN 6187 (field number JBM 2270), an adult male captured 4 October, 2020, by J. Brito and M. Herrera, preserved as dry skin, skull, postcranial skeleton, and muscle and liver biopsies in 95% ethanol.

Paratypes: MECN 3723, adult male, preserved as dry skin and cleaned skull, collected in Provincia de Zamora Chinchipe, Reserva Biológica Tapichalaca (−4.492083, −79.129778, 2,500 m) by F. Reid on 22 November, 2013. QCAZ 13175, adult male, preserved as dry skin and cleaned skull, collected in Provincia de Loja, La Libertad, Shucos (−3.82083, −79.1174619, 2,900 m) by S. Lobos on 28 January, 2012.

Type locality: Ecuador, Provincia de El Oro, Cantón Chilla, Ashigsho (−3.44785°, −79.61015°, WGS84 coordinates taken by GPS at the site of collection; elevation 2,539 m).

Etymology: Named in honor of Neisi Dajomes Barrera, an Ecuadorian athlete weightlifting athlete born in the Provincia de Pastaza; Ecuadorian female Olympic gold medalist. The species epithet is formed from the name “Neisi” taken as a noun in apposition.

Diagnosis: A species of Chilomys which can be identified by the following combination of characters: long nasal (~8.4–8.8 mm); zygomatic plate straight; M1 without anteromedian flexus; M1–M2 with indistinct mesoloph; M2 with narrowed hypoflexus (similar in width to mesoflexus); m1 without anteromedian flexus; hemal arches absent.

Morphological description of the holotype and variation: Small body size for the genus (head and body length combined range between 95 and 100 mm). Dark neutral gray (color 299) dorsal fur; short hairs (medium length on back = 6.5 mm) with dark neutral gray (color 299) base and olive-brown (color 278) tips. Dark neutral gray (color 299) venter coat, with hairs (medium length = 6.5 mm) with pale neutral gray (color 297) base and smoke gray (color 266) tips. Jet black (color 300) periocular ring. Postauricular patch present. Mystacial vibrissae long, thick at base and thin towards tip, exceeding ears when tilted backwards; superciliary vibrissae 1 present, genal vibrissae 1 present. Ears (15–16 mm from notch to margin) externally covered by short smoke gray (color 266) hairs, dark neutral gray (color 299) inner surface, pale neutral gray (color 296) margin. Narrow and ground cinnamon drab (color 259) metatarsal patch, which extends to base of phalanges; dorsal surface of foot with round scales and small interspaces. Plantar surface with six pads, including four interdigitals of similar size, thenar and hypothenar pads small and with large interspace; space between pads is smooth (Fig. 5F). Short digit I reaches base of digit II; digit II slightly smaller than digit III and digit III slightly smaller than digit IV; short digit V reaches middle of digit IV. Long tail (~128–136 mm; ~135% of HB), unicolor fawn (color 258) except for apex, which is white (up to 15–25 mm). Tail with 15–16 rows of scales per cm on axis; square scales with three hairs each, which extend over 1.5 rows of scales; naked-looking tail except for tip, where it presents a small brush of up to 4 mm.

Cranium small for the genus (~24.01–24.7 mm of CIL). Short and narrow rostrum, with nasal bones that extend to incisors; poorly developed gnathic process. Posterior margin of nasal bone does not exceed plane of lacrimal bone. Shallow zygomatic notch (deep in old specimen). Small and rounded lacrimal bones, almost entirely welded to maxillae. Wide interorbital region with smooth outer edges, exposing alveolar maxillary processes in dorsal view (Fig. 15). Supraorbital region with diverging posterior borders. Frontoparietal suture V-shaped. Broad rounded and inflated braincase, concave at outer edges. Developed ethmoturbinals (Fig. 13G). Wide zygomatic plaque, comparatively longer than length of M1, and posterior border reaches anterior face of M1. Zygomatic arches sturdy with jugals spanning a large segment of each mid-arch. Small supraorbital foramen with posterior border in line with M3 (Fig. 7D). Alisphenoid strut wide and robust. Carotid circulatory pattern type 3; carotid canal large, stapedial foramen small, without alisphenoid squamous groove and without sphenofrontal foramen. Subsquamous fenestra one third size of postglenoid foramen (Fig. 15); hamular process of squamosal thin and long, and distally applied on mastoid capsule. Slightly triangular tegmen tympanic, superimposed with suspensory process of squamous. Lateral expressions of parietals present; bullae small; pars flaccida of tympanic membrane present, large; orbicular apophysis of malleus well-developed. Paraoccipital process small. Hill foramen long (Fig. 14D); short and narrow incisive foramen with curved edges without reaching plane defined by anterior faces of M1; a pair of ridges on either side of palatine foramina and in front of M1. Premaxillary capsule widened, parallel-sided and narrow at rear ends; maxillary septum of incisive foramen slim and long. Long and wide palate, mesopterygoid fossa not reaching M3. Posterolateral palatal pit small. Wide mesopterygoid fossa, with by a medium palatal process present. Inconspicuous sphenopalatine vacuities covered by roof of palate. Basisphenoid narrow. Large D-shaped foramen ovale. Middle lacerate foramen narrow. Auditory bullae small and uninflated with large and wide eustachian tube (Fig. 8D).

Figure 15 Chilomys neisi sp. nov.

Chilomys neisi sp. nov. (Ashigsho, Chilla, El Oro, Ecuador): cranium in (A) dorsal, (B) ventral, and (C) lateral views, and mandible in (D) labial view (MECN 6187, holotype). Photographs by J. Brito.

Dentary short, with short and wide coronoid process (not extending beyond upper edge of condylar process); short and thin mental foramen.

Proodont upper incisors (Thomas angle of ~102°; Fig. 7D) with orange and smooth front enamel; crested and pentalophodont molars, with noticeably thick enamel. Maxillary molar rows parallel; main cusps opposite (Fig. 9E) and sloping backwards when viewed from side. M1 rectangular in outline without anteromedian flexus; thin and short anteroflexus; small anteroloph; indistinct mesoloph; reduced posteroloph; internal closure of mesoflexus ends in a fosseta. M2 squared in outline; mesoloph showing same condition as in M1; narrowed hypoflexus (distinctly narrower than mesoflexus); internal fosseta larger than fosseta of M1. M3 less than half the size of M2; M3 rounded in outline with conspicuous anteroloph; central fosseta large but smaller than M2. Lower molars with main cusps opposite (Fig. 9F) and sloping forwards when viewed from side. First lower molar (m1) without anteromedian flexus; large anterolabial cingulum; thin and short mesolophid. Mesoloph absent; noticeable anterolabial cingulum. Hypoflexid of m3 long and wide; hypoflexid well-developed and deep in m1–m3.

Tuberculum of first rib articulates with transverse processes of seventh cervical vertebra. First and second thoracic vertebrae have differentially elongated neural spine. Vertebral column is composed of 19 thoracicolumbar, 16th with moderately developed anapophyses and 17th with little developed anapophyses, four sacrals (fused), and 39 caudal vertebrae without hemal arches; 12 ribs. Scapular notch extends to half of scapula and scapular spine not reaching caudal border; supratrochlear foramen of humerus absent; contact between tibia and fibula occurs in more medial part of these bones and fibula reaches 55% of length of tibia.

Comparisons: Chilomys neisi sp. nov., is a small Chilomys species that inhabits Ecuador (Fig. 3; Table 4) and it can be confused with C. percequilloi sp. nov., but differs from C. percequilloi sp. nov. (states in parenthesis) by the following structures: Thomas angle ~102° (Thomas angle ~92°); M1 without anteromedian flexus (with anteromedian flexus); M1–M2 with indistinct mesoloph (present); M2 with narrowed hypoflexus, similar in width to mesoflexus (broader hypoflexus, similar in width to mesoflexus); m1 without anteromedian flexus (with anteromedian flexus); hemal arches absent (present).

Chilomys instans, another small species that inhabits the eastern flank of Ecuador (Fig. 9:see Table 4) could be confused with C. neisi sp. nov. However, it can be differentiated from Chilomys instans (states in parentheses) by the following structures: M1 without anteromedian flexus (with anteromedian flexus); M1–M2 indistinct mesoloph (distinct mesoloph); M2 with narrowed hypoflexus, distinctly narrower than mesoflexus (broader hypoflexus, similar in width to mesoflexus). A detailed comparison with all Chilomys species is presented in Table 3.

Distribution: Chilomys neisi sp. nov., has the southernmost distribution of all species described in this work; it is known from two locations in the Provincias de Zamora Chinchipe and El Oro, Ecuador (Fig. 10), at elevation around 2,500–2,900 m. To the north, C. neisi sp. nov. is recorded at Ashigsho, Chilla (Provincia de El Oro) at an elevation of 2,500 m; to the south, the species occurs at Reserva Tapichalaca (Provincia de Zamora Chinchipe) at an altitude of 2,900 m. The annual average temperature corresponds to 16.8 °C. The coldest times are reached in August in Tapichalaca (minimum temperature of 9.6 °C) and the warmest in September in Chilla (maximum temperature of 25.3 °C). Average precipitation is 1,075 mm per year, the driest month (July and August) being in Chilla (23 mm per month) and the wettest in March in Tapichalaca, 190 mm per month (Hijmans et al., 2005).

Natural history: The zoogeographic area where Chilomys neisi sp. nov., occurs is Temperate (Albuja et al., 2012). The ecosystem corresponds to the montane forest (Ministerio del Ambiente del Ecuador, 2013), which is characterized by trees with abundant orchids, ferns, and bromeliads. Chilomys neisi sp. nov., was collected in mature forest where the undergrowth is visually dominated by herbaceous families such as Poaceae (Chusquea sp.), Araceae, and Melastomataceae. On the steep slopes, the palm (Ceroxylon sp.) predominates. Stomach content from one specimen revealed Coleoptera (one larva), and Chrysomelidae (one adult). Chilomys neisi sp. nov., was collected in sympatry with the didelphids Marmosops caucae Caenolestes caniventer and C. condorensis, and the rodents Akodon mollis, Nephelomys albigularis, Microryzomys minutus, Oreoryzomys balneator, and Thomasomys taczanowskii.

Chilomys percequilloi sp. nov. Brito, Tinoco, García and Pardiñas

urn:lsid:zoobank.org:act:0985D3E1-87C6-4E2E-B95A-53FB0C1C81C2

Percequillo Forest Mouse, Ratón del bosque de Percequillo (in Spanish)

Holotype: MECN 5854 (field number JBM 1959), an adult male captured 26 January, 2018, by J. Brito, and N. Tinoco, preserved as dry skin, skull, postcranial skeleton and muscle and liver biopsies in 95% ethanol.

Paratopotypes: MECN 5822, adult female, preserved as dry skin and cleaned skull, collected by J. Brito, J. Curay and R. Garcia on 12 September, 2017. MECN 5858, and MECN 5859, adult males, QCAZ 17552, juvenile male, QCAZ 17555, and QCAZ 17557, adult males, all preserved as dry skins and cleaned skulls, collected by J. Brito and N. Tinoco on 29 January, 2018.

Paratypes: MEPN 6921, adult male, preserved as dry skin and cleaned skull, collected in Provincia de Napo, Laguna Guataloma (−0.28°, −78.13°, 4,000 m) by M. Cueva on 30 September, 1996. MEPN 5827, and MEPN 5828, juvenile males, preserved as dry skins and cleaned skulls, collected in Laguna Loreto (−03°, −78.15°, 4,050 m) by W. Pozo and F. Trujillo on 29 November, 1996. MEPN 10063, adult male, preserved as dry skin and cleaned skull, collected in Cuyuja (−0.402°, −78.018°, 2,775 m) by L. Albuja and F. Trujillo on 29 May, 2005. MEPN 9937, juvenile male, preserved as dry skin and cleaned skull, collected in Río Azuela (−0.75555°, −77.59083°, 1,600 m) by L. Albuja and F. Trujillo on 23 June, 2004. QCAZ 4189, male adult, preserved as dry skin and cleaned skull, collected in Papallacta (−0.33422°, −78.1455°, 3,570 m) by S. Burneo on 2 June, 2001. MECN 6338, juvenile male, and MECN 6361, adult male, preserved as dry skins and cleaned skulls, collected in Provincia de Tungurahua, Reserva Naturetrek Vizcaya (−1.35871°, −78.39558°, 2,391 m) by J. Brito, R. Vargas, E. Pilozo, T. Recalde and E. Peña on 13 May, 2021. MECN 6362, adult male, preserved as dry skin and cleaned skull, collected in Parque Nacional Llanganates (−1.355580°, −78.379180°, 3,268 m) by J. Brito, R. Vargas, E. Pilozo, T. Recalde and E. Peña on 18 May, 2021. MECN 3796, juvenile male, preserved as dry skin and cleaned skull, collected in Provincia de Morona Santiago, Sardinayacu, Parque Nacional Sangay (−2.074306°, −78.211833°, 1,766 m) by J. Brito, H. Orellana and G. Tenecota on 21 June, 2014. MECN 4327, MECN 4329, adult females, and MECN 4328, adult male, preserved as dry skins and cleaned skulls, collected in Cerro Sambalán, Parque Nacional Sangay (−2.206139°, −78.452694°, 2,851 m) by J. Brito, G. Pozo, and R. Ojala-Barbour on 15 January, 2015.

Type locality: Ecuador, Provincia de Morona Santiago, Cantón Méndez, Parroquia Patuca, Cordillera de Kutukú (−2.78722°, −78.13166°, WGS84 coordinates taken by GPS at the site of collection; elevation 2,215 m).

Etymology: This species is named in honor of Alexandre Reis Percequillo (nickname PC), Brazilian contemporary biologist devoted to the study of Neotropical mammal fauna and a specialist in oryzomyine rodents. The species epithet is formed from the surname “Percequillo,” taken as a noun in the genitive case, with the Latin suffix “i” (ICZN 31.1.2).

Diagnosis: A species of Chilomys identified by the following combination of characters: tail with 18–20 rows of scales per centimeter on axis; zygomatic plate sloping backwards; M1–M2 with mesoloph; M2 with broader hypoflexus (similar in width to mesoflexus); m1 with anteromedian flexus; hemal arches present.

Morphological description of the holotype and variation: Small body size for the genus (head and body length combined range between 76 and 90 mm). Light neutral gray (color 297) dorsal fur; short hairs (medium length on back = 7.1 mm) with dark neutral gray (color 299) base and smoke gray (color 266) tips. Smoke gray (color 155) ventral coat, with hairs (medium length = 5.5 mm) with pale neutral gray (color 297) base and smoke gray (color 266) tips. Surface of throat and chest lighter than rest of belly. Jet black (color 300) periocular ring (Fig. 16). Postauricular patch absent. Mystacial vibrissae long, thick at base and thin towards tip, exceeding shoulder when tilted backwards; superciliary vibrissae 1 present, genal vibrissae 1 present. Ears (14–17 mm from notch to margin) externally covered by short smoke gray (color 266) hairs, pale buff (color 1) inner surface, medium neutral gray (color 298) margin.

Figure 16 External aspect of Chilomys percequilloi sp. nov.

External aspect of Chilomys percequilloi sp. nov. (MECN 5854, holotype), an adult male from Cordillera de Kutukú, Morona Santiago, Ecuador. Photograph by J. Brito.

Narrow and ground cinnamon drab (color 259) metatarsal patch, which extends to base of phalanges. Plantar surface with six pads, including four interdigitals of similar size, hypothenar pad smaller than thenar pad and with space between them; space between pads is covered by scales (Fig. 5H). Short digit I reaches base of digit II; digit II slightly smaller than digit III and digit III same size as digit IV; short digit V (apparently somewhat opposable) reaches middle of digit IV. Long tail (118–133 mm; ~146% of HB), unicolor fawn (color 258) (in some specimens ventral tail is slightly paler than back) except for apex, which is white (up to 15 mm). Tail with 18–20 rows of scales per cm on axis; rectangular scales with three hairs each, which extend over 1.5 rows of scales in dorsal basal sector; naked-looking tail except for tip, where it presents a small brush of up to 5 mm. Protruding anus. Three mammary pairs in females, one pectoral, one abdominal, one inguinal; females with a long white clitoris (~5 mm) that contrasts with color of belly.

Cranium small for the genus (22.8–24.3 mm of CIL). Short and narrow rostrum, with nasal bones that do not extend to incisors; poorly developed gnathic process (Fig. 17). Posterior margin of nasal bone does not exceed plane of lacrimal bone. Shallow zygomatic notch. Small and rounded lacrimal bones. Wide interorbital region with smooth outer edges, exposing alveolar maxillary processes in dorsal view. Supraorbital region with diverging posterior borders. Frontoparietal suture U-shaped. Broad, rounded and inflated braincase, concave at outer edges. Developed ethmoturbinals (Fig. 13H). Wide zygomatic plaque, comparatively longer than length of M1, and with posterior edge not reaching maxillary row. Zygomatic arches sturdy with jugals spanning a large segment of each mid-arch. Large supraorbital foramen with posterior border in line with M3 mesoflexus (Fig. 9G). Alisphenoid strut wide and robust, thin and delicate (MECN 3796) or absent (MECN 4327) in young individuals. Carotid circulatory pattern type 3; carotid canal large, stapedial foramen small, without alisphenoid squamous groove and without sphenofrontal foramen. Subsquamous fenestra twice smaller than postglenoid foramen (Fig. 17); hamular process of squamosal thin and long, and distally applied on mastoid capsule. Slightly triangular tegmen tympanic, superimposed with suspensory process of squamous. Lateral expressions of parietals present; bullae small; pars flaccida of tympanic membrane present, large; orbicular apophysis of malleus well-developed. Paraoccipital process small. Hill foramen small; short and narrow incisive foramen with curved edges without reaching plane defined by anterior faces of M1. Premaxillary capsule widened, parallel-sided and narrow at rear ends; maxillary septum of incisive foramen slim and long. Long and wide palate. Posterolateral palatal pit small. Narrow mesopterygoid fossa produced by a medium process of short and blunt palatine. Inconspicuous sphenopalatine vacuities covered by roof of palate. Large D-shaped foramen ovale. Middle lacerate foramen narrow. Auditory bullae small and uninflated with large and narrow eustachian tube (Fig. 8E).

Figure 17 Chilomys percequilloi sp. nov.

Chilomys percequilloi sp. nov. (Cordillera de Kutukú, Morona Santiago, Ecuador): cranium in (A) dorsal, (B) ventral, and (C) lateral views, and mandible in (D) labial view (MECN 5854, holotype). Three-dimensional reconstruction by C. Koch and J. Brito.

Dentary short, with short and narrow coronoid process (not extending beyond upper edge of condylar process); short and thin mental foramen of jaw.

Proodont upper incisors (Thomas angle ~92°; Fig. 7E) with orange and smooth front enamel; crested and pentalophodont molars, with noticeably thick enamel. Maxillary molar rows slightly convergent backwards and slightly hypsodont; coronal surfaces crested; main cusps opposite (Fig. 9H) and sloping backwards when viewed from side. M1 rectangular in outline with procingulum divided by anteromedian flexus into subequal anterolabial and anterolingual conules (in young specimens); deep anteroflexus (in young specimens); short and wide anteroloph; slim and long mesoloph; reduced posteroloph; internal closure of mesoflexus ends in a fosseta. M2 squared in outline; mesoloph showing same condition as in M1; broader hypoflexus (similar in width to mesoflexus); internal fosseta larger than fosseta of M1. M3 less than half the size of M2; M3 rounded in outline with conspicuous anteroloph; central fosseta large but smaller than M2. Lower molars with main cusps opposite (Fig. 9H) and sloping forwards when viewed from side; tip of incisors is sharp. First lower molar (m1) with anteromedian flexid inconspicuous that divides procingulum into subequal anterolabial and anterolingual conulids; large anterolabial cingulum; ectolophid present; thin and short mesolophid. Mesoloph of m2 showing same condition as in m1; ectostylid present; noticeable anterolabial cingulum; hypoflexid of m3 long and wide. Hypoflexid well-developed and deep in m1–m3.

Tuberculum of first rib articulates with transverse processes of seventh cervical vertebra. First and second thoracic vertebrae have differentially elongated neural spine. Vertebral column is composed of 19 thoracicolumbar, 16th with moderately developed anapophyses and 17th with little developed anapophyses, four sacrals (fused), and 36–40 caudal vertebrae; with complete hemal arches in second and third caudal vertebra; 12 ribs.

Comparisons: Chilomys percequilloi sp. nov., is one of the small Chilomys species that inhabits Ecuador (Fig. 3; Table 4). However, this species could be confused with C. neisi sp. nov. (states in parenthesis), but can be differentiated by the following traits: Thomas angle ~92° (Thomas angle ~102°); M1 with anteromedian flexus (without anteromedian flexus); M1–M2 with distinct mesoloph (indistinct mesoloph); M2 with broader hypoflexus, similar in width to mesoflexus (narrow hypoflexus, similar in width to mesoflexus); m1 with anteromedian flexus (without anteromedian flexus); hemal arches present (hemal arches absent).

Another species that inhabits the eastern flank of Ecuador (Fig. 10) which is of similar size (see Table 4) and could be confused with Chilomys percequilloi sp. nov., is Chilomys instans. However, the former can be differentiated from C. instans (states in parenthesis) by the following traits: Thomas angle ~92° (Thomas angle ~100°); m1 with anteromedian flexus (without anteromedian flexus); hemal arches present (hemal arches absent). A detailed comparison with all species of Chilomys is presented in Table 3.

Distribution: Known from several localities in the provinces of Napo to Morona Santiago (Ecuador), on the eastern flank of the Andes (Fig. 10), at an elevation between 1,600 to 4,050 m. Chilomys percequilloi sp. nov., has the widest range of the five species described in the present work, covering about 300 lineal kilometres between its northernmost (Azuela River, Provincia de Sucumbíos, MEPN 9937) and southernmost records (Cordillera de Kutukú, Provincia de Morona Santiago, MECN 5858). Likewise, it has the highest altitudinal range of all known species of Chilomys, since it is distributed in the eastern foothills of the Andes, in the Provincias de Sucumbíos, Napo, Tungurahua and Morona Santiago, although one would expect to find it also in the provinces of Cotopaxi and Chimborazo. Taking into account this altitudinal range we can suppose that Chilomys percequilloi sp. nov., has an ample tolerance to different environmental conditions. The average temperature among the recording localities is 13.3 °C, with a significant variation between 1.1 °C as the minimum temperature in November for Loreto Lagoon (Provincia de Napo), to 25.6 °C as the maximum temperature in November for Kutukú (Provincia de Morona Santiago). With respect to the precipitation, the annual average is 1,960 mm, also showing a significant variation ranging from 1,090 mm at Reserva Naturetrek Vizcaya (Provincia de Tungurahua) to 3,400 mm in Kutukú (Hijmans et al., 2005).

Natural history: The zoogeographic area where C. percequilloi sp. nov., occurs is Eastern Sub-Tropical, Temperate and Altoandino (Albuja et al., 2012). The ecosystem corresponds to the montane forest (Ministerio del Ambiente del Ecuador, 2013), which is characterized by trees with abundant orchids, ferns, and bromeliads. Chilomys percequilloi sp. nov., was collected in mature forest where the undergrowth is visually dominated by herbaceous families such as Poaceae (Chusquea sp.), Araceae, and Melastomataceae. On the steep slopes, the royal palm (Dictyocaryum lamarckianum) predominates. Stomach contents of three specimens were analysed. Identifiable prey items were 25% Lepidoptera, 25% Blattodea, 25% Diptera, and 25% Acari (Supplemental S4). Chilomys percequilloi sp. nov., was collected in sympatry with the didelphids Marmosa germana, Marmosops caucae and Monodelphis adusta, and the rodents Akodon aerosus, A. mollis, Nephelomys auriventer, N. nimbosus, Oreoryzomys balneator, Rhipidomys albujai, Thomasomys pardignasi, T. cinnameus, T. erro, and T. salazari.

Chilomys weksleri sp. nov. Brito, García, Pinto and Pardiñas

urn:lsid:zoobank.org:act:292D0BA6-BF28-4C0D-BF26-1433DE9AE423

Weksler Forest Mouse, Ratón del bosque de Weksler (in Spanish)

Holotype: MECN 6365, an adult female captured 5 October, 2020, by C. Nivelo and J. Viera, preserved as skull and carcass in ethanol, and muscle and liver biopsies in 95% ethanol.

Paratopotypes: MECN 6363, juvenile male, and MECN 6364, adult female, preserved as cleaned skulls and carcasses in ethanol, collected 1 October, 2020, by C. Nivelo and J. Vieira.

Paratypes: MEPN 9954, adult male, preserved as dry skin and cleaned skull, collected in Provincia de Pichincha, Mindo Nambillo (−0,051°, −78.54°, 2,600 m) by M. Cueva on 30 September, 1996. MECN 4925, adult male, preserved as dry skin and cleaned skull, collected in Reserva Geobotánica Pululahua (0.02025°, −78.493138°, 3,190 m) by J. Curay, J. Brito, R. Vargas and K. Valdivieso on 2 April, 2016. MECN 4171, adult male, preserved as dry skin and cleaned skull, collected in Hacienda Tambillo Alto (−0.4073917°, −78.565991°, 2,833 m) by R. García on 9 November, 2014. QCAZ 1787, adult male, preserved as ethanol, collected by P. Jarrín on 9 September, 1996. QCAZ 8693, QCAZ 8694, and QCAZ 8695, (sex and age indeterminate), preserved as cleaned skulls and carcasses in ethanol, collected in Provincia de Cotopaxi, Reserva Integral Otonga (−0.4189°, −79.0039°, 2,000 m) by K. Helgen and C. M. Pinto on 15 August, 2006.

Type locality: Ecuador, Provincia de Cotopaxi, Cantón Sigchos, Parroquia San Francisco de Las Pampas, Reserva Integral Otonga (−0.685367°, −78.995089°, WGS84 coordinates taken by GPS at the site of collection; elevation 1,654 m).

Etymology: This species is named in honor of Marcelo Weksler, Brazilian contemporary biologist devoted to the study of living and fossil Neotropical cricetids. The species epithet is formed from the surname “Weksler,” taken as a noun in the genitive case, with the Latin suffix “i” (ICZN 31.1.2).

Diagnosis: A species of Chilomys which can be identified by the following combination of characters: Head and body length ~74–85 mm; tail longer than head and body length combined (~143–153%); dorsal surface of foot with round scales and small interspaces; zygomatic plate leaning forward; M2 with broader hypoflexus (similar in width to mesoflexus); m1 with anteromedian flexus.

Morphological description of the holotype and variation: Small body size (head and body length combined range between 74 and 85 mm). Brown (color 277) dorsal fur; short hairs (medium length on back = 6.5 mm), dark neutral gray (color 299). Dark neutral gray (color 299) venter coat, with hairs (medium length = 6.5 mm) with pale neutral gray (color 297) base and smoke gray (color 266) tips. Jet black (color 300) periocular ring. Postauricular patch absent. Mystacial vibrissae long, thick at base and thin towards tip, exceeding ears when tilted backwards; superciliary vibrissae 1 present, genal vibrissae 1 present. Ears (~14–16 mm from notch to margin) externally covered by short smoke gray (color 266) hairs, with whitish inner surface, and pale neutral gray (color 296) margin.

Narrow and sayal brown (color 41) metatarsal patch, which extends to base of phalanges; dorsal surface of foot with round scales and small interspaces. Plantar surface with six pads, including four interdigitals of similar size, thenar and hypothenar pads large and with small interspace; space between pads is smooth (Fig. 5J). Short digit I reaches base of digit II; digit II slightly smaller than digit III and digit III slightly smaller than digit IV; short digit V reaches middle of digit IV. Whitish unguals equal or slightly surpassing tip of claws. Long tail (103–121 mm; ~148% of HB), buff (color 15) and bicolor (dark above and whitish below) except for apex, which is white (up to 22.6 mm). Tail with ~22–23 rows of scales per cm on axis; square scales with three hairs each, which extend over 1–1.5 rows of scales; naked-looking tail except for tip, where it presents a small brush of up to 3.5 mm.

Cranium small for the genus (~21.5–23.5 mm of CIL). Short and narrow rostrum, with nasal bones that extend to incisors; poorly developed gnathic process. Posterior margin of nasal bone does not reach plane of lacrimal bone. Shallow zygomatic notch. Small and elongated lacrimal bones, almost entirely welded to maxillae. Wide interorbital region with smooth outer edges, without exposing alveolar maxillary processes in dorsal view (Fig. 18A). Supraorbital region with diverging posterior borders. Frontoparietal suture V-shaped. Broad rounded and inflated braincase, concave at outer edges. Wide zygomatic plaque, comparatively longer than length of M1, and with posterior edge not reaching maxillary row. Zygomatic arches sturdy with jugals spanning a large segment of each mid-arch. Large supraorbital foramen with posterior border in line with M2 (Fig. 7F). Alisphenoid strut present. Carotid circulatory pattern type 3; carotid canal large, stapedial foramen small, without alisphenoid squamous groove and without sphenofrontal foramen. Subsquamous fenestra three times smaller than postglenoid foramen (Fig. 18C); hamular process of squamosal thin and long, and in contact on mastoid capsule. Slightly triangular tegmen tympanic, superimposed with suspensory process of squamous. Lateral expressions of parietals present; bullae small; pars flaccida of tympanic membrane present; orbicular apophysis of malleus well-developed. Paraoccipital process small. Hill foramen small (Fig. 14F); short and narrow incisive foramen with curved edges not reaching plane defined by anterior faces of M1; a pair of ridges on either side of palatine foramina and in front of M1. Premaxillary capsule widened, converging and narrow at rear ends; maxillary septum of incisive foramen slim and long. Long and wide palate, with mesopterygoid fossa not reaching M3. Posterolateral palatal pit small. Wide mesopterygoid fossa, with by a medium palatal process present. Inconspicuous sphenopalatine vacuities covered by roof of palate. Basisphenoid narrow. Large D-shaped foramen ovale. Middle lacerate foramen very narrow. Auditory bullae small and slightly inflated with short and wide eustachian tube (Fig. 8F).

Figure 18 Chilomys weksleri sp. nov.

Chilomys weksleri sp. nov. (Reserva Intergral Otonga, Cotopaxi, Ecuador): cranium in (A) dorsal, (B) ventral, and (C) lateral views, and mandible in (D) labial view (MECN 6365, holotype). Photographs by J. Brito.

Dentary short, with short and narrow coronoid process (not extending beyond upper edge of condylar process); elongated and thin mental foramen.

Proodont upper incisors (Thomas angle of ~92°; Fig. 7F) with orange and smooth front enamel; crested and pentalophodont molars, with noticeably thick enamel. Maxillary molar rows parallel; main cusps opposite (Fig. 9I) and sloping backwards when viewed from side. M1 rectangular in outline with procingulum divided by anteromedian flexus into subequal anterolabial and anterolingual conules; thin and short anteroflexus; long paraflexus; long anteroloph; long and thin mesoloph; reduced posteroloph; internal closure of mesoflexus ends in a fosseta; anterior mure long. M2 squared in outline; mesoloph showing same condition as in M1; broader hypoflexus (similar in width to mesoflexus); internal fosseta similar to fosseta of M1; long and thin protoflexus. M3 less than half the size of M2; M3 rounded in outline with conspicuous anteroloph; central fosseta large, similar to M2. Lower molars with main cusps opposite (Fig. 9J) and sloping forwards when viewed from side. First lower molar (m1) with anteromedian flexus that divides procingulum into subequal anterolabial and anterolingual conulids; short and thin mesolophid; small anterolabial cingulum; ectostylid present. Mesolophid of m2 showing same condition as in m1; ectostylid present. Hypoflexid of m3 long and wide; hypoflexid well-developed and deep in m1–m3.

Comparisons: Chilomys weksleri sp. nov., that the smallest species of the genus, inhabits Ecuador (Fig. 3; Table 4). However, it could be confused in the first instance with C. georgeledecii sp. nov., (states in parenthesis), from which can be differentiated by the following traits: Thomas angle ~102° (Thomas angle ~92°); zygomatic plate comparatively wider than M1 and leaning forward (comparatively similar to M1 and slightly tilted backwards); M2 with broader hypoflexus, similar in width to mesoflexus (narrowed hypoflexus, but distinctly narrower than mesoflexus); m1 with anteromedian flexus (without anteromedian flexus). See detailed comparison of all Chilomys species is presented in Table 3.

Distribution: Chilomys weksleri sp. nov., is distributed in the foothills of the Western Cordillera of the central Andes of Ecuador, between the Provincias de Pichincha and Cotopaxi (Fig. 10), at elevations between 1,600 and 3,200 m. The four recorded localities register a temperature average of 14.2 °C with the greatest fluctuation between annual minimum and maximum temperatures in August, reaching a minimum temperature of 5.2 °C and a maximum temperature of 23.6 °C. The average precipitation is 1,500 mm per year, with the lowest monthly precipitation of 28 mm in July, and the highest precipitation of 225 mm occurring in March (Hijmans et al., 2005).

Natural history: The zoogeographic area where Chilomys weksleri sp. nov., occurs is Temperate (Albuja et al., 2012). The ecosystem corresponds to the montane forest (Ministerio del Ambiente del Ecuador, 2013), which is characterized by trees with abundant orchids, ferns, and bromeliads. Chilomys weksleri sp. nov., was collected in mature forest where the undergrowth is visually dominated by herbaceous families such as Poaceae (Chusquea sp.), Araceae, and Melastomataceae. The species was collected in sympatry with the didelphids Marmosops caucae, Caenolestes caniventer and C. fuliginosus, and the rodents Akodon mollis, Nephelomys moerex, Microryzomys minutus, Thomasomys aureus, T. baeops, and T. silvestris.

Discussion

Diagnosing Chilomys

Most of the history of the knowledge of Chilomys reflects the scarcity of specimens available for study. Thomas (1895, 1897) described the type species and the genus based on one skull, which was also studied by Ellerman (1941: 372). Osgood (1912) erected C. fumeus based on two individuals. With the second decade of the past century the collection surveys developed by several American institutions in Ecuador, Colombia and Venezuela retrieved an important input of specimens. Especially, the field efforts of George H. H. Tate (during 1920–1924 in Ecuador), Philip Hershkovitz (around 1950 in Colombia), and Charles Handley (around 1969 in Venezuela) greatly enriched the number of collected Chilomys (e.g., https://collections-zoology.fieldmuseum.org/list?search_fulltext=Chilomys; Handley, 1976). In any case, the new material did not change the poor perception on this Andean form, which was reduced to a monotypic condition after the influential treatises of Gyldenstolpe (1932) and Cabrera (1961). As was summarized by Voss (2003: 23) “The morphologically distinctive genus Chilomys Thomas (1897) is currently thought to contain only a single valid species, C. instans (Thomas, 1895); another nominal taxon, C. fumeus Osgood (1912) is either a subspecies or synonym according to Cabrera (1961) and Musser & Carleton (1993)… A revision of this long-neglected northern-Andean endemic genus is necessary…”

Pacheco (2015a) deserves the merit to having produced the first generic description of Chilomys including aspects of mostly external (based on the examination of dry skins) and cranial morphology. Interesting to note, despite more than a century, Chilomys remained explicitly undiagnosed; when Thomas (1897) coined the generic the name, probably he judged enough the description already provided for instans advanced few years before (Thomas, 1895). From a formal point of view, therefore, Chilomys lacks a diagnosis, although the list of generic features collated by Gyldenstolpe (1932: 37) can be considered as such. We provide an improved diagnosis and present several morphological features for the first time.

Family Cricetidae Fischer, 1817

Subfamily Sigmodontinae Wagner, 1843

Tribe Thomasomyini Steadman and Ray, 1982

Genus Chilomys Thomas, 1897

Type species (by monotypy).—Oryzomys instans Thomas, 1895.

Etymology.—None originally, but Néstor Cazzaniga (in litteris) suggested that Thomas (1897) employed the Greek noun τιλός (chilos), meaning “grass” to distinguish Chilomys from Oryzomys, whose generic epithet is composed of ὄρσζα (oryza), meaning “rice.”

Geographic distribution.—Known from Andean montane forests and Páramo-forest ecotone from northwestern Venezuela in the north to northern Perú in the south, generally ranging between 1,000 and 4,050 m above sea level.

Chronological distribution.—Recent; no fossils are known.

Contents.—The type species (C. instans) and, in order of nomination, C. fumeus Osgood, 1912, C. carapazi sp. nov. Brito & Pardiñas, C. georgeledecii sp. nov. Brito, Tinoco, García, Koch & Pardiñas, C. neisi sp. nov. Brito, Tinoco, García, Koch, & Pardiñas, C. percequilloi sp. nov. Brito, Tinoco, García & Pardiñas, and C. weksleri sp. nov. Brito, García, Pinto & Pardiñas (this paper).

Emended diagnosis.—Small-bodied (head and body length ~85 mm; body weight ~18 grams; condylobasal length ~23 mm), long-tailed (~140% of head and body length) thomasomyines distinguished by the following combination of characters: Fur soft and straight, dark gray to gray-brown with venter not countershaded; ears medium sized; eyes small and rimmed by very short hair; mystacial vibrissae numerous and somewhat rigid, typically not surpassing auricular pinna when pressed backwards; pollex very small; hindfeet narrow and relatively long, dorsally unusually scaly and scarcely haired and ventrally typically smooth and having six well-defined pads; pes claws medium sized, moderately hooked and basally covered by whitish ungual tufts; protuberant but not prominent anus; large clitoris; tail unicolored thinly haired and typically having fleshy-colored distal inch; three pairs of mammae in inguinal, abdominal, and thoracic positions; cranium with markedly domed profile with comparatively short rostrum and large, rounded, deep braincase; nasals narrow, shorter than premaxillae, evenly converging backwards; shallow zygomatic notches; interorbital region broad and smooth; coronal suture U or V-shaped; large interparietal; nasolacrimal capsules inflated; zygomatic plates narrow and high; zygomatic arches robust but not broadened and with their ventral margins placed distinctly above orbital floor; jugals long; large supraorbital foramina; conspicuous lateral expression of parietals; carotid circulation representing pattern 3 (sensu Voss, 1988); alisphenoid struts typically present; tegmen tympanic overlaps suspensory processes of squamosals; hamular processes of squamosals large and distally applied to well-developed mastoid capsules; dorsal aperture of ectotympanic ring open; gnathic process absent; medium and large Hill foramen; incisive foramina short distantly placed from first upper molars; posterior parts of upper diastema marked by swollen ridges lying on either sides of incisive foramina, combined with masseteric scars distinctly placed anterior to root of zygomatic plates; palate broad, uncomplicated and typically long; parapterygoid plates well-defined and large, perforated by conspicuous ovale foramina and transverse canals; hamular processes of pterygoids large; otic bullae flasked-shaped; upper incisors ungrooved and markedly proodont exhibiting Thomas’ angles between 92–102° and in some species clearly visible in front of nasals in a vertical view of skull; molars noticeably small (microdont condition) and brachydont with thick enamel; mesolophs present in M1–M2 but tending to disappear with wear, closed remnants of mesoflexus persistent as fossettes in M1–M3; M3 markedly smaller than M2; m3 sigmoid-shaped; lower incisors slender and pointed; lower diastema flat and almost horizontal but markedly broad; lower border of dentary tending to flat; coronoid processes well-developed and hooked; condyles broad; capsular process well-developed; angular processes short; stomach unilocular-hemiglandular with subequal distribution of cornified and glandular epithelia; caecum small and single; gall bladder present; baculum with thin and sinoid-curved shaft and deeply concave and narrow base; complex penis with lateral cartilaginous digits thick and pointed and medial digit slim and blunter; one pair of preputials and larger medial and ventral prostates than lateral ones (after Thomas, 1895; Osgood, 1912; Gyldenstolpe, 1932; Ellerman, 1941; Carleton, 1973; Steppan, 1995; Voss, 1991; Pacheco, 2015a; Calderón-Capote et al., 2016; this paper).

Description.—Pacheco (2015a) provided a description of external and cranial features of Chilomys. We have elaborated here the anatomical fields that have been scarcely explored or not mentioned so far. Externally, Chilomys is characterized by a prominent head in comparison to body, with eyes of large size, beautifully rimmed in black and magnified by periocular rings of very short hairs. Mystacial vibrissae are numerous, blackish and whitish, of moderate length (not surpassing posterior margin of pinna when pressed against body) and inserted in a partially naked field, extended to both sides of nose and confluent with periocular ring (Fig. 19A). Chilomys has a simple rhinarium characterized by a naked and broad dorsal integumental fold, non-sculptured nasal pads with almost undiscernibly horizontal grooves, ventral integumental folds flanking lower external angle, nostrils of moderate size and a marked median sulcus. Both upper and lower lips are covered by short hairs (Fig. 19B). Ears are noticeable although partially hidden in dorsal fur, appearing naked but covered by very short hairs. Pinnae are pinkish due to subjacent blood irrigation and characterized by a well-developed antihelix and antitragus and closed up by marked crura of antihelix delimiting a patent fossa triangularis and a recessed concha (Fig. 19A).

Figure 19 Selected external and soft anatomical traits of Chilomys.

Selected external and soft anatomical traits of Chilomys: (A) external aspect of an individual in wild (C. georgeledecii sp. nov; MECN 5381, paratype); (B) rhinarium in anterior view (C. georgeledecii sp. nov.; MECN 6205, paratype); (C) palmar surface of right fore foot (C. georgeledecii sp. nov.; MECN 5381, paratype); (D) urogenital region (C. georgeledecii sp. nov.; MECN 5381, paratype); (E) soft palate (C. weksleri sp. nov.; MECN 6364, paratopotype); and (F) tongue in dorsal view (C. weksleri sp. nov.; MECN 6364, paratopotype). Abbreviations: 1–5, digits; a, anus; ah, antihelix; at, antitragus; c, clitoris; ch, concha; cv, circumvallate papilla; d1–d3, diastemal rugae; ft, fossa triangularis; he, helix; if, lower integumental fold; i1–15, interdental rugae; my, mystacial vibrissae; n, nostril; np, nasal pad; pe, periocular ring; ph, philtrum; su, semilunar sulcus; uf, upper integumental fold; v, vagina. Photographs by J. Brito.

Osgood (1912: 53) emphasized the unusually scaly nature of the dorsal surface of the pes in C. fumeus, a common condition in other species of Chilomys (Figs. 5A–5I). With a minor degree of variation, shortness and scarcity of hairs in upperparts of fore and hind foot contribute to make evident scales. The latter are subrectangular in shape, disposed in tight rows or sometimes appearing as disordered and extending scaly lining to fingers (i.e., first and second phalanx are dorsally covered by about eight rows of scales). In addition, another peculiarity of cheiridia scales is the coloration, described as “…being dark colored with lighter margins” (Osgood, 1912: 53), a characteristic not seen in dry skins but very vivid in fresh or ethanol-preserver specimens. Finally, each digit is apically embellished with a dense tuft of ungual vibrissae, mostly expressed over third phalanx and scarcely reaching end of claw. The latter condition is highlighted by the hallux, whose claw seems to be naked. The tip of each finger is covered basally with a turgid, deep callus and distally with an acute, moderately broad, medium-length, and ventrally open claw. The sole of the hind foot has been described as lacking imbrications, a statement coined by Osgood (1912) and quoted by contemporary authors (see Pacheco, 2015a). Surely Osgood (1912) emphasized the naked nature of the undersurface, without scales but having a varyingly number of “granules” (a term used herein to denote and expand interspaces between interdigital pads and between first interdigital and thenar pads). Both digit and metatarsal pads are bulging, rounded and roughly subequal in size and shape, even the hypothenar, which is typically smaller than the other plantar structures. Regarding the manus, Osgood (1912: 53) highlighted the minute condition of the pollex, only an excrescence related to the thenar pad (Fig. 19C). The remainder four digits are subequal in length, markedly stocky and ringed, ending in deep calluses and topped off with broadened, but hooked claws. Almost the entire palm is occupied by the pads, being the interspaces smooth but crossed by a visible stria separating the digital group from the palmar group.

A protuberant anus is not prominent, according to Pacheco (2015b), but appears as a noticeable orifice produced at the top of a fleshy bulbous structure (Fig. 19D). The clitoris is decidedly large (Fig. 19D), well haired and whitish and the mammae are disposed in three pairs, including an inguinal one.

Although the cranium of Chilomys was characterized by a short rostrum and the development of the braincase strongly dominated, the most impressive characteristic of the genus is clearly its microdonty. Viewed from below, the diastemal portion is capable of containing twice the molar series. More indeed, the diastemic palate anterior to the incisive foramina shows almost the same length as the latter mentioned structures and bears a well-enlarged Hill foramen. An additional unusual trait with occurrence in the diastemal portion was described by Osgood (1912: 54) as “… a pair of swollen ridges lying on either side of the palatine foramina and in front of Ml” (Fig. 14). These ridges, plus marked “scars” for the origin of the masseter superficialis and serrated premaxillary-maxillary sutures conform a set of characteristics presumably associated to a powerful masticatory musculature. Despite being described as narrow (Pacheco, 2015a), the zygomatic plate is a solid and tall structure, with a short free upper border and far from the degenerative type that characterizes several small Andean sigmodontines (Thomas, 1927). This is in line with robust zygomatic arches and well-developed jugals. Even having a long palate (according to the definition of Hershkovitz, 1962), the parapterygoid region is also impressive in Chilomys, and the same can be said for the unusually large pterygoids. Finally, the braincase is firmly globular, giving the cranium a noticeable depth in lateral view, as there is no perceptible basicranial deflection. Several bony structures associated to this region appear magnified such as the infraorbital foramen, the hamular process of the squamosal, the dorsally produced tegmen tympani, etc.

The soft diastemal palate is very ample, and surpasses the interdental palate in this respect. It is crossed by three entire rugae, which are well separated from each other (Fig. 19E). Five interdental rugae complete the dotation of the soft palate being partially bowed and well divided middorsally by a perceptible longitudinal sulcus. It is interesting to note how fixed the number of diastemic ridges appears to be, since three, if the anterior most bearing the incisive papilla is counted as one (sensu Quay, 1954: fig. 1), are widespread in cricetids. Thus, even the diastemal palate is large in Chilomys, there is no enlargement of the ridges but an enlargement of the rugae interspaces. The tongue fills the mouth closely when the molars are near in occlusion. Its length comprises three times the molar series and its width is not greatly enlarged in the distal portion with respect to the intermolar portion. A shallow median sulcus dissects the dorsal surface of the distal 1/3 of the tongue, and an indistinct semilunar sulcus defines the anterior limit of the torus linguae. From the apex to a short distance anterior to the epiglottis, the surface is lined with filiform papillae resembling horny denticles. A single circumvallate papilla is located on the dorsal midline of the tongue, shortly anterior to the epiglottis (Fig. 19F).

The stomach gross morphology was previously assessed based on three specimens of C. weksleri sp. nov., from Pichincha, Ecuador and typified as unilocular-hemiglandular (Carleton, 1973: fig. 3A, mentioned as C. instans). We here reaffirm this characterization after the dissection of more than 10 individuals representing several of the described species. No morphological differences attributable to taxonomy have been detected. A uniform unilocular-hemiglandular pattern with roughly equivalent distribution of cornified and glandular epithelia was registered; the walls of the corpus are thin and the internal surface moderately smooth to the naked eye, while the antrum has thicker walls; the bordering fold looks like a thick cord and probably acts effectively to produce a functional bicamerality in this kind of the unilocular stomach; very close to the esophageal opening the bordering fold bends strongly to the right, forming a narrow and definitive esophageal channel; finally, the stomach has a well-defined and broadened prepyloric part (Fig. 20). We also confirm the widespread occurrence of a gallbladder in Chilomys (detected in C. georgeledecii sp. nov. MECN 5381, 5387, 6303, 6315, 6337, 6364; C. instans MECN 4769; C. percequilloi sp. nov. MECN 5593, 6338; C. neisi sp. nov. MECN 6187; and C. weksleri sp. nov. MECN 4171, 6365), which was first reported by Voss (1991: table 4) based on five specimens originally assigned to C. instans (AMNH 63370-63372; UMMZ 155619, 155620). In four species (C. georgeledecii sp. nov., C. percequilloi sp. nov., C. neisi sp. nov., and C. weksleri sp. nov.) examined to assert the general morphology of the intestine, the post-caecum portion was noticeably short (about 40 mm), while the pre-caecum intestine accounted for a medium length of about 180 mm. The gross morphology of the caecum was subequal among the species studied, consisting mostly of a single sac with two main constrictions, no appendix, and a rather simple colonic region (Fig. 21). This general configuration is consistent with a fiber-free, enriched-protein diet (Vorontsov, 1982).

Figure 20 Gross morphology of the stomach in two species of Chilomys.

Gross morphology of the stomach in two species of Chilomys: (A, C) ventral external and (B, D) internal views in C. georgeledecii sp. nov. (A, B; MECN 6337, paratype); and in C. percequilloi sp. nov. (MECN 6338, paratype). Abbreviations: b, bordering fold; co, cornified epithelium; d, duodenum; ge, glandular epithelium; i, incisura angularis. Photographs by J. Brito.

Figure 21 External views of the partial digestive system in several species of Chilomys.

External views of the partial digestive system in several species of Chilomys: (A) C. georgeledecii sp. nov. (MECN 6337, paratype); (B) C. neisi sp. nov. (MECN 6187, holotype); (C) C. percequilloi sp. nov. (MECN 6338, paratype); and (D) C. weksleri sp. nov. (MECN 6365, holotype). Photographs by J. Brito.

Very little has been reported about the postcranial skeleton of Chilomys (see Steppan, 1995). The tuberculum of the first rib articulates with the transverse processes of the seventh cervical; the first thoracic and the second thoracic vertebra have differentially elongated neural spine. The remainder portion of the axial skeleton is composed of 19 thoracicolumbar, the 16th with moderately developed anapophyses and the 17th with little developed anapophyses, four sacrals (fused), and 36–40 caudal vertebrae with/without hemal arches. The recorded number of ribs is 12. The scapular notch extends to half of the scapula and the scapular spine does not reach the caudal border. A cursory inspection of the main long bones did not show the supratrochlear foramen of the humerus. The contact between the tibia and fibula occurs in the more medial part of these bones and the fibula reaches 50–60% of the length of the tibia.

Dental key traits: incisor procumbency and microdonty

After more than a century, and despite the considerable diversity added to the universe of sigmodontine rodents, Thomas’s (1895) keen perception of the incisor procumbency in Chilomys is sustained (Fig. 7). He stated “Upper incisors unusually thrown forwards, so that in a vertical view of the skull they are clearly visible in front of the nasals” (Thomas, 1895: 369). This condition was later termed as proodont (or pro-odont) by the same author (Thomas, 1919), who also typified orthodont and opisthodont to describe angular variations of the upper (unusually also applied to lower) incisors in rodents. To avoid any confusion, this author has illustrated how to take the angle formed by the upper incisor (Thomas, 1919: fig. 1), a descriptor today known as the “angle of Thomas” and employed for taxonomic differentiation (e.g., Myers, 1989). Interesting to note, Hershkovitz (1962: 101–102) adopted Thomas’s terminology (and meaning) but introduced different vertical and horizontal planes to assess the procumbency of the incisors. While Thomas (1919) measured the angle formed by the chord of the incisor arc against the molar plane, Hershkovitz (1962: fig. 19) used the interception between a “vertical incisive-plane” with the “basal-incisive plane.” Although the latter was clearly defined (Hershkovitz, 1962: fig. 21), the former was illustrated but not described; intuitively, the “incisive-plane” should be the plane crossing the upper incisors by the centroid of the exposed (i.e., extralveolar) portion (Hershkovitz, 1962: fig. 21). More recent authors followed mostly the definitions of Hershkovitz (1962) (e.g., Steppan, 1995, Pacheco, 2003) but also introduced subtle variations in how planes are defined and/or interpreted. For example, Weksler (2006: 43) explained “The degree of upper incisor procumbency is defined by the position of the cutting edge of the incisor relative to the vertical-incisive plane (Hershkovitz, 1962; Steppan, 1995).” Therefore, this author introduced a new element compared to Thomas (1919), the cutting edge, and eliminated one of the planes employed by Hershkovitz (1962), the “basal-incisive plane.” In this context, it is clear that no angle can be calculated because a plane is missing, and the perception of incisor orientation is limited to a more or less subjective appreciation of the way these dents protrude forward or not. Although the original proposition of Thomas (1919) was criticized (see discussion in Akersten, 1973), it still seems to be the most objective way to assess incisor procumbency independently if the character is scrutinized for phylogenetical scoring or taxonomical/functional interpretation.

A cursory revision of the 90 living genera included within Sigmodontinae is conclusive that the widespread condition is opisthodonty (including the extreme state called hyper-opisthodont; see Steppan, 1995: 17). Orthodonty is much less frequent and proodonty is extremely rare. Regarding the latter two conditions, genus assignments varied between different authors, probably due to anarchy in estimating procumbency (vide supra). As such, Ellerman (1941) designated some species of Necromys (Akodontini), two genera of Phyllotini (Auliscomys and Galenomys), one Oryzomyini (Scolomys), and Chilomys as proodont, the latter being designated as strongly proodont. Hershkovitz (1962) agrees with the proodont condition of Galenomys, but limits the case of Auliscomys to the species boliviensis. However, Steppan (1995: 19) explicitly opposed the conclusion of Hershkovitz’s (1962) and typified both genera as orthodont, thus eradicating the proodonty of Phyllotini and the associated high-crowned sigmodontines (i.e., Andinomyini, Euneomyini, and Reithrodontini). Weksler (2006: 43) did the same, but with respect to Oryzomyini, apart from the impossibility of finding perpendicularity between two vertical planes, he concluded that “Amphinectomys, Handleyomys, Melanomys, Oryzomys [= Mindomys] hammondi, Scolomys, and Sigmodontomys [= Tanyuromys] aphrastus have orthodont incisors with the cutting edge perpendicular to the vertical-incisive plane.” Pacheco (2003) restricted proodonty to Abrawayaomys, Chilomys, and a few species of Thomasomys, while Pardiñas, Teta & D’Elía (2009: table 2) showed variation in upper incisor angles in Abrawayaomys, implying a transition from opisthodontic to proodontic conditions. More recently, Teta et al. (2017) indicated orthodont or slightly proodont upper incisors in Abrothrix and Chelemys.

Apparently, extreme proodonty is not exclusive to Chilomys, as envisioned by Thomas (1895), but it clearly deserves attention because this trait distinguishes the genus among the thomasomyines. The orientation of the upper incisors takes on a new meaning when combined with another dental characteristic of Chilomys: microdonty. The latter is applied here according to Schmidt-Kittler (2006), implying that the molar tooth-row is comparatively short judged against the entire skull length. No previous authors mentioned this condition for Chilomys, but it is evident based on a direct inspection of the materials, or the ratio obtained dividing condyle-incisive length/upper molars tooth-row length (Table 4). Micro- and macrodonty are virtually unexplored traits of sigmodontines (Ronez et al., 2020). At least intuitively, however, the diminution in molar size can be linked to insectivory, a relationship that has been discussed and demonstrated for other groups of mammals (e.g., Freeman, 2000; Ungar, 2010). There is no published information on the diet of Chilomys, although the genus has been categorized as insectivorous (Maestri et al., 2017: appendix S1). The examined stomach contents of the specimens dissected here exclusively revealed insect and other invertebrate (Supplemental S4) remains including entire animals or large pieces of exoskeletons. This highly animal-protein diet is in agreement with the single caecum morphology displayed by Chilomys (Vorontsov, 1982; Fig. 21). Having all these elements at hand, we can advance the hypothesis that this thomasomyine is an invertebrate-eater and its diet triggered two dental characteristics already discussed: proodonty and microdonty. Partially in contrast to a more widespread phenotype in the morphological evolution of sigmodontines that involves the “long-nosed” condition associated to insectivory (see Martinez et al., 2018; Missagia & Perini, 2018; Pardiñas et al., 2021), specialization in short-rostrum Chilomys is a privilege of incisor procumbency. It is not known if this thomasomyine uses these teeth to pick and/or pinch invertebrates, or if they serve as digging tools when foraging. Two additional dental traits deserve mention here, as they are probably related to both non-exclusive strategies. Thomas (1895) described the lower incisors of C. instans as long and very slender, and we can confirm this characterization, but also emphasizing their acute tips (in fact, we pricked our fingers several times while working with the mandibles during). In addition, the enamel of the molars of Chilomys looks unusually thick considering their minute size. In the context of the brachydont condition of the genus, the thickening of the enamel can be interpreted as a positively selected trait to counteract excessive wear caused by ingestion of soil particles (Madden, 2015).

Facing hidden Andean diversity in cricetids

The speciose condition of Chilomys is not necessarily surprising. This thomasomyine genus is widespread in northern Andes and covers more than 10 degrees of latitude from western Venezuela to northern Perú (Medina et al., 2016). As a typical inhabitant of the montane forest belt that developed on both Andean slopes, Chilomys is not only exposed to the selection pressure of the moderate ecological gradient imposed by altitudinal variation (roughly from 1,600 to 4,050 m), but its range is also strongly fragmented by mountain discontinuity and fluvial systems (e.g., Táchira Depression, Huancabamba Depression, Mira and Jubones rivers). If we add to this current context the historical dimension, all the necessary elements are present to favor active speciation.

Our knowledge of the real diversity of Chilomys is still incipient. There is virtually no data for huge portions of its range, including almost all Colombian and Peruvian populations, but also for the southern Ecuadorian Andes. Therefore, nothing solid can be said about the history of diversification of the genus. However, focusing on the Ecuadorian diversity sampled, one could probably propose an allopatric speciation model (e.g., Patton & Smith, 1992). Time estimates derived from molecular phylogenies, although probably biased by poor fossil control, suggest the Chilomys originated in the Pleistocene (i.e., no older than 2.5 MA) and is considered sister to another rare thomasomyine, Aepeomys (see Schenk & Steppan, 2018). Chilomys would have been exposed to numerous contractions and expansions of Andean vegetation belts triggered by the impact of glacial-interglacial cycles. Even admitting the high degree of regional variability, the multivariate local conditions, the occurrence of non-analogue vegetational assemblages, and likely volcanic events, etc., from the pioneering studies to the recent most contributions on Quaternary paleoecology a clear picture emerges: montane forests have been fragmented, compressed, expanded, and/or isolated many times (e.g., Van der Hammen, 1974; Marchant, Boom & Hooghiemstra, 2002; Bakker, Moscol Olivera & Hooghiemstra, 2008; Cárdenas et al., 2011; Loughlin Nicholas et al., 2018). Classical palynological long-term profiles, such as those of the High Plain of Bogota (western side of the Cordillera Oriental in Colombia), point to several replacements between Páramo-type vegetation and Andean forests during most of the Plio-Pleistocene (e.g., Clapperton, 1993 and the references cited therein). We are convinced that the diversification of Chilomys was in part the result of Pleistocene expansion and contraction cycles that led to geographic isolation and/or secondary contact of species, as has been suggested for several other Andean animal and plant species (e.g., Rull, 2011; Nevado et al., 2018).

Numerous cricetids occurring in northern Andes are currently treated as mono- or paucispecific genera (Patton, Pardiñas & D’Elía, 2015). These are, among others, members of the tribes Ichthyomyini (e.g., Neusticomys), Oryzomyini (e.g., Microryzomys, Oreoryzomys), Neomicroxini (e.g., Neomicroxus), etc. They share large geographical ranges with Chilomys and are exposed to a variety of environmental gradients and topographical discontinuities. More indeed, preliminary studies published or not, are revealing unexpected geographical variability. Hence, Chilomys is surely not a unique case of an Andean sigmodontine with hidden diversity. An important degree of genetic variation, partially coincident with different geographic Andean units was recently reported for populations traditionally referred to Neomicroxus latebricola, a Páramo sigmodontine (Cañón et al., 2020). Ongoing research is revealing that Oreoryzomys, supposedly monotypic and even a plausible synonym of Microryzomys (see Carleton & Genus Microryzomys Thomas, 2017), is not only a valid genus, but also consists of at least three species (J. Brito et al., 2021, unpublished data). Coupled with the extensive sampling being done by several teams of scientists (e.g., Instituto Nacional de Biodiversidad, Pontificia Universidad Católica del Ecuador) and the refinement of molecular studies and other kind of approaches, it is not unlikely that numerous species will be described or resurrected from nominal forms during this decade.

We began this section by stating that this addition to the specific diversity of Chilomys is not surprising, but challenging, and we would like to end this contribution with a brief elaboration on this second aspect of our findings. The impact of hidden diversity in several fields of our comprehension of the evolutionary biology is a candent topic (see Richter et al., 2021 and the references cited therein). Thomasomyines are probably one of the most remarkable expressions of the sigmodontine radiation in Andean habitats. However, their convoluted history as a tribe (e.g., mixed with the oryzomyines for many years), their supposed moderate diversity, and a perceptible stasis in their study, have led to a poor participation when the evolution of the subfamily is addressed (e.g., Parada, D’Elía & Palma, 2015; Schenk & Steppan, 2018). Extirpating Rhagomys, incorporated to Thomasomyini by D’Elía et al. (2006) but removed by Pardiñas et al. (2022), the tribe is currently composed of the living genera Aepeomys, Chilomys, Rhipidomys and Thomasomys, and the extinct Megaoryzomys being also in question of its tribal affiliation (Ronez et al., 2020). To date, our knowledge of Aepeomys and Chilomys is so scarce that attention to this suprageneric group has focused almost exclusively on Rhipidomys and Thomasomys. Although the latter is the most diverse living sigmodontine, with at least 47 species recognized as valid (Brito et al., 2021; Ruelas & Pacheco, 2021), this is not enough to make a clear impact in evolutionary explorations, since much of this diversity is not represented in molecular phylogenies (e.g., the most extensive contributions cover <35% of the species, see Brito et al., 2021; Ruelas & Pacheco, 2021). In addition, and judged generically, is a fact that the diversity of Thomasomyini is pale in comparison with even minor groups such as Abrotrichini or Ichthyomyini. Reached to this point we are persuaded that thomasomyines represent a suitable example of the negative effect of hidden diversity. After the present contribution, the diversity of Chilomys is raised to seven species and therefore the genus now integrates the group of those with moderate specific richness (between five to ten species, e.g., Abrothrix, Eligmodontia, Necromys, Nephelomys). However, the real issue is whether Thomasomys does not represent a complex of genera, as strongly suggested by the morphological and molecular data collected by several scholars (e.g., Pacheco, 2003; Voss, 2003). The division of Thomasomys into eight genera, the number of species groups proposed by Pacheco (2015b) and refined in subsequent studies (Brito et al., 2019; Brito et al., 2021), probably seems to cause over splitting. However, this scenario is not very different to the division of Oryzomys in several units of generic rank, as was proposed by Weksler (2006) and widely accepted (e.g., Patton, Pardiñas & D’Elía, 2015). There is still much to learn about the radiation of the thomasomyines, and unraveling their systematics is crucial to illuminating Andean biotic evolution and the history of the entire subfamily.

Conclusions

After more than a century of stasis in alpha taxonomy an integrative approach supported by extensive field sampling reveals that the poorly-known Andean thomasomyine Chilomys instans constitutes a complex of species. Five new species are described here, from Ecuadorian populations inhabiting montane forests on both sides of the Andes. Preliminarily, the newly revealed diversity can be attributed to allopatric speciation associated with the effect of Quaternary glacial-interglacial cycles on vegetation belts. Chilomys emerges as a morphologically distinctive Andean thomasomyine that exhibits unique specializations related to the procumbency of the incisors and probably associated to an invertebrate feeding strategy.

Supplemental Information

Supplemental Information 1 DNA Extraction Protocols.

Click here for additional data file.

Supplemental Information 2 Sequences used in the analysis genetic.

Click here for additional data file.

Supplemental Information 3 Comparison of dorsal, ventral and lateral views of the skull and occlusal views of the upper and lower row between Chilomys species (old individuals).

(A) C. carapazi sp. nov. (MECN 5291, holotype); (B) C. neisi sp. nov. (MECN 3723, paratype); (C) C. percequilloi sp. nov. (MECN 4328, paratype); (D) C. weksleri sp. nov. (MEPN 6036, paratype).

Click here for additional data file.

Supplemental Information 4 Stomach content composition of Chilomys.

Click here for additional data file.

Supplemental Information 5 Bayesian Inference phylogenetic tree based on the Cytochrome b-Cytb gene (800-1140 bp).

Click here for additional data file.

Supplemental Information 6 Bayesian Inference phylogenetic tree based on the Cytochrome Oxidase I - COI gene (600 bp).

Click here for additional data file.

Supplemental Information 7 Sequences used for genetic analysis.

Click here for additional data file.

To J. Robayo, J.P. Reyes, L. Jost, and H. Schneider of Basel Botanical Garden and Rainforest Trust; to the graduate biologists J. Curay, R. Vargas, C. Bravo, S. Pozo, K. Cuji, Z. Villacis, J. Guaya, J. Castro and E. Pilozo (the ‘Minion’ team), and G. Pozo, R. Ojala-Barbour and M. Herrera for invaluable field assistance; to the rangers of EcoMinga, especially H. Yela, T. Recalde, E. Peña, R. Peña, and M. Canticus for their deep efforts with field logistics; to D. Inclán and F. Prieto of INABIO, for their assistance in executing the work with local people and permanent support, to Roberto Portela Miguez (NHMUK) for access to collections and type material. To the Prefectura de El Oro for their support for sampling in Chilla. M. Herrera (INABIO) kindly collaborated by identifying the stomach content. We are deeply indebted to the above-mentioned persons and institutions. This is an Initiative Vorontsov 2030’s contribution # 2.

Appendix

APPENDIX 1 Studied specimens belong to the following mammal collections: FMNH, Field Museum of Natural History, USA; MECN, Instituto Nacional de Biodiversidad, Quito, Ecuador; NHMUK, Natural History Museum, London, United Kingdom; MEPN, Museo de la Escuela Politécnica Nacional, Quito, Ecuador; QCAZ, Museo Pontificia Universidad Católica del Ecuador, Quito, Ecuador; specimens marked with an * are holotypes.

Chilomys carapazi (n = 1): Ecuador, Provincia de Carchi, Reserva Drácula, Gualpi Km 18 (MECN 5291*).	
Chilomys fumeus (n = 1): Colombia, Cundinamarca, Páramo de Tamá (FMNH 18690*).	
Chilomys georgeledecii (n = 47): Ecuador, Provincia de Carchi, Reserva Drácula, Cerro Oscuro (MECN 4751–52, 4761); Gualpi Km 14 (MECN 4983, 4992–97); Gualpi Km 18 (MECN 4955–56, 4967–68, 4971, 5299, 5300–03, 5324, 5387, 5381, 5940, 5941, 5942); Guapilal km 12 (MECN 6181, 6205, 5923, 5926, 5921, 5924, 6032, 6033); Peñas Blancas-Pailón (MECN 5356, 5360, 5362, 5355, 5359, 5361, 6024*); Bosque La Esperanza (MECN 6303, 6323, 6315, 6327, 6337).	
Chilomys instans (n = 8): Colombia, Huila, San Agustín, Río Magdalena (FMNH 71498); Cundinamarca, Bogotá, Plains of Bogotá (NHMUK 1895.10.14.1*); San Cristóbal (FMNH 71629). Ecuador, Provincia de Carchi, Bosque de Polylepis Lodge (MECN 10875); Reserva Ecológica El Ángel (QCAZ 11188–91); Provincia de Imbabura, Palahuco (MECN 4769).	
Chilomys percequilloi (n = 26): Ecuador, Provincia de Napo, Oyacachi (MEPN 6921); Laguna Loreto (MEPN 5828); Cuyuja (MEPN 10063); Río Azuela, El Reventador (MEPN 9937); Papallacta (QCAZ 4107, 4154, 4188–89, 4194, 6253, MECN 6139); Provincia de Tungurahua, Reserva Naturetrek Vizcaya (MECN 6096–97, 6103–05); Reserva Naturetrek Candelaria (MECN 5593); Provincia de Morona Santiago, Parque Nacional Sangay, Sardinayacu (MECN 3796); Cerro Sambalán (MECN 4327–29); Kutukú (MECN 5822, 5830, 5854*, 5858–59).	
Chilomys neisi (n = 2): Ecuador, Provincia de Zamora Chinchipe, Tapichalaca (MECN 3723); Provincia de El Oro, Chilla, Ashigsho (MECN 6187*).	
Chilomys weksleri (n = 13): Ecuador, Provincia de Pichincha (NHMUK 1934.9.10.203–203A, 1954.555); Guarumos (MEPN 9954); Reserva Geobotánica Pululahua, Moraspungo (MECN 4925); Hacienda Tambillo Alto (MECN 4171); Provincia de Cotopaxi, Bosque Integral Otonga (QCAZ 1787, 8693–95, MECN 6363–64, 6365*).	

Additional Information and Declarations

Competing Interests

Author Contributions

Animal Ethics

Field Study Permissions

DNA Deposition

Data Availability

New Species Registration

The authors declare that they have no competing interests.

Jorge Brito conceived and designed the experiments, performed the experiments, analyzed the data, prepared figures and/or tables, authored or reviewed drafts of the paper, acquired the funds, and approved the final draft.

Nicolás Tinoco performed the experiments, analyzed the data, prepared figures and/or tables, authored or reviewed drafts of the paper, and approved the final draft.

C. Miguel Pinto performed the experiments, analyzed the data, prepared figures and/or tables, authored or reviewed drafts of the paper, and approved the final draft.

Rubí García analyzed the data, authored or reviewed drafts of the paper, and approved the final draft.

Claudia Koch performed the experiments, analyzed the data, authored or reviewed drafts of the paper, and approved the final draft.

Vincent Fernandez analyzed the data, authored or reviewed drafts of the paper, and approved the final draft.

Santiago Burneo analyzed the data, prepared figures and/or tables, authored or reviewed drafts of the paper, and approved the final draft.

Ulyses F. J. Pardiñas conceived and designed the experiments, performed the experiments, analyzed the data, prepared figures and/or tables, authored or reviewed drafts of the paper, and approved the final draft.

The following information was supplied relating to ethical approvals (i.e., approving body and any reference numbers):

Handling and all activities regarding specimens followed care and use ethical procedures recommended by the American Society of Mammalogists (Sikes, 2016). For the use and care of animals, we follow the guidelines of the Ministerio del Ambiente, Agua y Transición Ecológica del Ecuador, through scientific research authorization No 006-2015-IC-FLOFAU-DPAC MAE, No 003-2019-IC-FLO-FAU-DPAC/MAE, MAE-DNB-CM-2019-0126, and MAAE-ARSFC-2020-0642.

The following information was supplied relating to field study approvals (i.e., approving body and any reference numbers):

Field experiments were approved by the Ministerio del Ambiente, Agua y Transición Ecológica del Ecuador (permissions number No 006-2015-IC-FLO-FAU-DPAC/MAE, No 003-2019-ICFLO-FAU-DPAC/MAE), MAE-DNB-CM-2019-0126, and MAAE-ARSFC-2020-0642).

The following information was supplied regarding the deposition of DNA sequences:

The GenBank accession numbers, Cytb and COI sequences, and the specimens/material described and reviewed are available in the Supplemental File.

The Cytb and COI sequences are available at GenBank: OM703407–OM703427 and OM703428–OM703441.

The following information was supplied regarding data availability:

The sequences and location of the specimens studied in this work are available in the Supplemental File. All the examined specimens are detailed in Appendix 1.

The reconstructed CT data are available at MorphoBank Project DOI: 10.7934/P4152, http://dx.doi.org/10.7934/P4152.

Direct links to the data of the specimens used are as follows:

-Chilomys georgeledecii sp. nov. (MECN 6024, holotype):

cranium https://morphobank.org/index.php/Projects/Media/id/834883/project_id/4152;

mandible https://morphobank.org/index.php/Projects/Media/id/834884/project_id/4152.

-Chilomys instans (NHMUK 1895.10.14.1, holotype):

cranium https://morphobank.org/index.php/Projects/Media/id/834887/project_id/4152;

mandible https://morphobank.org/index.php/Projects/Media/id/834888/project_id/4152.

-Chilomys neisi sp. nov. (MECN 3726, paratype):

cranium https://morphobank.org/index.php/Projects/Media/id/834886/project_id/4152;

mandible https://morphobank.org/index.php/Projects/Media/id/834885/project_id/4152.

-Chilomys percequilloi sp. nov. (MECN 5854, holotype):

cranium https://morphobank.org/index.php/Projects/Media/id/834881/project_id/4152;

mandible https://morphobank.org/index.php/Projects/Media/id/834880/project_id/4152.

The following information was supplied regarding the registration of a newly described species:

Publication LSID: urn:lsid:zoobank.org:pub:22604A8F-0472-43EB-8D9F-9503C7AE4419

Species name:

Chilomys carapazi sp. nov. LSID: urn:lsid:zoobank.org:act:A12AF0E7-4465-4A9F-99B0-7E09DBDD5BBA

Chilomys georgeledecii sp. nov. LSID: urn:lsid:zoobank.org:act:BDEFF98C-5ED9-4DC7-8EC9-6ADE8BB297C1

Chilomys neisi sp. nov. LSID: urn:lsid:zoobank.org:act:F31C845C-DED1-4579-992D-9602FF14ADA6

Chilomys percequilloi sp. nov. LSID: urn:lsid:zoobank.org:act:0985D3E1-87C6-4E2E-B95A-53FB0C1C81C2

Chilomys weksleri sp. nov. LSID: urn:lsid:zoobank.org:act:292D0BA6-BF28-4C0D-BF26-1433DE9AE423.

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
