# Peer review of "Unlocking Andean sigmodontine diversity: five new species of Chilomys (Rodentia: Cricetidae) from the montane forests of Ecuador"

_PeerJ, doi:10.7717/peerj.13211_

## Round 0.1 · original submission · Major Revisions

Please address suggestions made by reviewers, especially reviewer 2, and provide a revised version of your manuscript.

·

Basic reporting

This study is written very well. The authors went to great efforts to assure that it is grammatically correct. It is easily understandable by all readers. The introduction is thorough; its details welcome. The procedures and methods of the study are laid out nicely. Figures are clear, tables are thorough, references are complete. Correct the tittle of Appendix 1 ('studied' specimens). This study is self-explanatory and self-explained.

Experimental design

I like this paper. It lays out the research objective clearly and reports on relevant information, in this case demonstrating diversity within Chilomys leading to diagnosis of new species, The standards of research are quite high and rigorous. Relevant ethical and scientific guidelines are acknowledged. Methods are eminently reproducible and testable.

Validity of the findings

This is a highly original study, completely appropriate and an advance for the field of research. The results are a useful, positive contribution. The conclusions flow from the observations and are well-summarized; future work is briefly outlined. Relevant data are presented in detail in the main article and in supplementary files.

Additional comments

This paper is done well and I appreciate the efforts of the authors to assure readability for an international audience. I have marked the pdf with edits that I hope are helpful and do not change meaning. Some are meant to help the narrative, sometimes by shortening it. Authors should check the preferred spelling of 'procumbency' and use it consistently.

·

Basic reporting

The manuscript is clear and technically correct. In general, the article includes a sufficient background about the taxonomy of the group under study. However, I consider that it is necessary to emphasize the phylogenetic relationships of the genus with other genera of Sigmodontinae rodents, as that would help to understand the sampling of external groups in phylogenetic analyses. Furthermore, the Introduction should specify why this study is restricted almost exclusively to Ecuador, even though there are populations in Venezuela, Colombia, and Peru. In this sense, I think that the information indicated in the first paragraph of the results should be raised in some way in the introduction. Thus, it would be possible to know the scope and limitations of this taxonomic study. I also recommend following a more logical order in the presentation of evidence throughout the entire manuscript. If the authors decide to start the materials and methods with the morphological part, then I think that this should be presented at the beginning of the results (and not the molecular results).

Experimental design

Morphology
• It is not completely clear what is the purpose of the X-ray Micro CT. Is it used as a source of evidence to distinguish potentially new species? Or, once the lineages were detected, is it used to better characterize those new species? In addition, it should be explained why this process was performed only for three specimens (holotypes). On the other hand, the author uses two multivariate analyses (PCA and DFA), which are very common in taxonomic studies. However, sometimes they are used without solid justification. I highly recommend specifying what the objective of each of the two analyzes is and thus justify their use. See, for example, Strauss, R. E. (2010). (pp. 73-91). Springer, Berlin, Heidelberg.

Genetics
• It is not clear how many specimens were sequenced. Following up on my comments about multivariate analyses, Cytb and COI are commonly used in mammalian taxonomic studies. I think the authors should at least briefly mention why they use both and not just one, considering they are mitochondrial. I also recommend specifying the PCR protocols used for this study, especially if the authors are using a combination of two protocols (i.e. Bonvicino & Moreira, 2001; Smith & Patton, 1999).
• The phylogenetic analyses should be better explained. Hoy many sequences were included in the analyses? What was the length of the amplified fragments per gene and what was the result of the assembly? Is there missing data? Were the same analyses done for each gene separately and also for the two genes put together? Looking at the supplemental material, I see the author used Rhagomys, Thomasomys, and Rhipidomys as external groups, but that should be clearly mentioned and justified in the text. In the discussion, the authors mention that Aepeomys is sister to Chilomys; then, why this genus was not included as an external group?
• The authors mention that they also used sequences from GenBank. Please, specify what other sequences you mean. Other sequences of the genus Chilomys? Sequences of external groups? Sequences from other studies?
• It should be specified whether the genetic distance analysis was done for each gene separately or both concatenated.

Validity of the findings

• The author mentions that Chilomys is “embedded in a clade with Rhipidomys, and Thomasomys”. Although this is true, I think this finding is obvious since they were the only genera used as outgroups ((and rooted with Rhagomys). The Materials and Methods should contain the rationale for why these genera were selected as outgroups. Because the results for the Cytb and COI genes vary as to the relationship of Chilomys to the Rhipidomys and Thomasomys, I think the authors could extend the outgroup sampling to discuss the monophyly of Chilomys and its sister group. If it is not possible to expand this sampling to carry out new analyses, then I recommend that the authors expand the discussion around the possible uncertainty of the relationship of Chilomys with other genera.
• Regarding the analysis of genetic distances, I think the authors only present the result for the Cytb (Table 1). Why is the COI result not presented?
• In my opinion, the results from the two multivariate analyses are poorly explained and underutilized. How is the variation distributed among the samples? What are the variables that most contribute to this variation? Are there variables that help discriminate some species more from others? Why is a classification analysis not shown for the samples?
• Perhaps my biggest comment is regarding the decision to consider 5 new species. It is not clear how the authors integrate the results from the phylogenetic analyses, the species delimitation methods, and the morphometric analyses. For example, the Cytb phylogeny seems to support five lineages, but the COI maybe three or four. The delimitation method even suggests many more than five species. The morphometric study is uncertain since the authors seem to use the five groups identified by the phylogenetic analysis of Cytb as a priori lineages. I recommend adding a section in the results section that integrates the evidence and thus justifies the recognition of 5 species. Overall, I suspect that the authors decide to base their conclusions on the phylogenetic tree derived from the Cytb sequences. I am not saying that it is a wrong decision, but that it must be clear and well justified based on all the evidence that the authors present.
• Given the sampling (practically restricted to Ecuador), I highly recommend that the authors mention the limitations in the discovery and description of these 5 species for Ecuador. In addition, what are the limitations of the genus diagnosis if samples from other countries are not reviewed in detail?
• The discussion about the diversification of the genus Chilomys is very interesting, but I think it is outside the scope of this study and appears to be speculative. Therefore, I suggest that that part be removed from the present manuscript.

Additional comments

I suggest that the list of examined specimens be treated as Appendix and not as supplementary material. I think the specimens examined are the most important part of taxonomic reviews and should not be relegated to supplementary material.

Reviewer 3 ·

Basic reporting

.

Experimental design

.

Validity of the findings

.

Additional comments

I am sending to you my suggestions and appreciation of the manuscript entitled “Unlocking Andean sigmodontine diversity: Five new species of Chilomys (Rodentia: Cricetidae) from the montane forests of Ecuador” (#68489).

The work done by Dr. Brito and his collaborators is clear, complete and follows the rules and norms of the journal. Authors followed the criteria and the high quality of the figures and tables, and citations and literature are impeccable. The manuscript is clearly written in professional and unambiguous language.
I consider the authors made excellent fieldwork efforts, and after many years of research in scientific collections, they compiled an extensive dataset that includes external and internal morphological data of the different species here proposed, including information obtained from x-rays of the skull, which makes this work a reference for the proposal of new mammal species.
However, the molecular analyses are not as complete as the morphological ones (it was not included any nuclear marker), but all other evidence is trustworthy, including the support of the different mitochondrial clades.
I only disagree in the recognition of the clade Chilomys carapazi sp. Nov. Brito and Pardiñas as an independent species, and I would recommend discarding it as a new species for the following reasons. After analyzing the diagnosis of the species and the information obtained throughout the analysis of this single specimen, I found the proposal very risky, since there is no molecular evidence that supports it, and the morphological analysis was based is an old specimen with a wearing teeth pattern that does not correspond to the ages of the rest of the individuals (Fig. 8). In other words, even though the PCA analysis showed this specimen was different, the comparison with other specimens is not clear (and possibly not valid because authors could be just comparing specimens of different ages). The lack of strong differentiation of that specimen resulted in its recognition as new species based on the combination of multiples characters, which in my opinion is not enough to recognized it as a different species.

Annotated reviews are not available for download in order to protect the identity of reviewers who chose to remain anonymous.

---

## Round 0.2 · accepted · Accept

The authors have provided a revised version of the manuscript, addressing most of the concerns of the reviewers. In a few cases where the authors chose not to follow the reviewers' suggestions, adequate explanations are provided. Thus, I consider it appropriate to accept the revised manuscript as it is and not send it to a new round of reviews.